# The myocardium utilizes a platelet-derived growth factor receptor alpha (Pdgfra)–phosphoinositide 3-kinase (PI3K) signaling cascade to steer toward the midline during zebrafish heart tube formation

Rabina Shrestha, Tess McCann, Harini Saravanan, Jaret Lieberth, Prashanna Koirala, Joshua Bloomekatz*

Department of Biology, University of Mississippi, University, United States

**\*For correspondence:**
josh@olemiss.edu

**Competing interest:** The authors declare that no competing interests exist.

**Abstract** Coordinated cell movement is a fundamental process in organ formation. During heart development, bilateral myocardial precursors collectively move toward the midline (cardiac fusion) to form the primitive heart tube. Extrinsic influences such as the adjacent anterior endoderm are known to be required for cardiac fusion. We previously showed however, that the platelet-derived growth factor receptor alpha (Pdgfra) is also required for cardiac fusion (Bloomekatz et al., 2017). Nevertheless, an intrinsic mechanism that regulates myocardial movement has not been elucidated. Here, we show that the phosphoinositide 3-kinase (PI3K) intracellular signaling pathway has an essential intrinsic role in the myocardium directing movement toward the midline. In vivo imaging further reveals midline-oriented dynamic myocardial membrane protrusions that become unpolarized in PI3K-inhibited zebrafish embryos where myocardial movements are misdirected and slower. Moreover, we find that PI3K activity is dependent on and interacts with Pdgfra to regulate myocardial movement. Together our findings reveal an intrinsic myocardial steering mechanism that responds to extrinsic cues during the initiation of cardiac development.

## Editor's evaluation

This is a valuable study that shows the involvement of phosphoinositide 3-kinase (PI3K) signaling downstream of platelet-derived growth factor receptor α in latero-medial migration of cardiomyocytes during the formation of the early heart tube during zebrafish development. The authors provide convincing evidence for the role of PI3K in cardiomyocyte migration using multiple PI3K inhibitory drugs, expression of a dominant negative PI3K subunit, and rescue of the Pdgfaa ligand over-expression phenotype using mild PI3K inhibition, approaches which show strong alignment and which are quantified using live imaging. The demonstration of cardiomyocyte protrusions biased in the direction of migration, and randomised after PI3K inhibition, is a promising area for future exploration.

## Introduction

During organogenesis, cell progenitor populations often need to move from their origin of specification to a new location in order to form a functional organ. Deficient or inappropriate movement can

underlie congenital defects and disease. Directing these movements can involve extrinsic factors such as chemical and mechanical cues from neighboring tissues and the local environment as well as intrinsic mechanisms such as intracellular signaling and polarized protrusions (*Schumacher, 2019*). Progenitor cell movement occurs during cardiac development, where myocardial cells are specified bilaterally on either side of the embryo (*Stainier et al., 1993*). To form a single heart that is centrally located, these bilateral populations must move to the midline and merge (*Wilens, 1955*; *Rawles, 1936*). As they move, myocardial cells undergo a mesenchymal-to-epithelial (MET) transition forming intercellular junctions and subsequently moving together as an epithelial collective (*Trinh and Stainier, 2004*; *Jackson et al., 2017*; *Dominguez et al., 2023*; *Linask, 1992*; *Holtzman et al., 2007*). This process is known as cardiac fusion and occurs in all vertebrates (*Davidson et al., 2005*; *Evans et al., 2010*).

External influence from the adjacent endoderm is essential for the collective movement of myocardial cells toward the midline. Mutations in zebrafish and mice which inhibit endoderm specification or disrupt endoderm morphogenesis result in cardia bifida – a phenotype in which the bilateral myocardial populations fail to merge (*Holtzman et al., 2007*; *Alexander et al., 1999*; *Kupperman et al., 2000*; *Osborne et al., 2008*; *Kawahara et al., 2009*; *Kikuchi et al., 2001*; *Mendelson et al., 2015*; *Molkentin et al., 1997*; *Ye and Lin, 2013*; *Li et al., 2004*; *Yelon et al., 1999*). Similar phenotypes also occur in chicks and rats when the endoderm is mechanically disrupted (*Goss, 1935*; *Rosenquist, 1970*; *Varner and Taber, 2012*). Studies simultaneously observing endoderm and myocardial movement have found a correlation between the movements of these two tissues, suggesting a model in which the endoderm provides the mechanical force that pulls myocardial cells toward the midline (*Varner and Taber, 2012*; *Cui et al., 2009*; *Ye et al., 2015*; *Aleksandrova et al., 2015*). Yet, these correlations do not occur at all stages of cardiac fusion, indicating that myocardial cells may also use intrinsic mechanisms to actively move toward the midline. Indeed, recent studies revealing a role for the receptor tyrosine kinase, platelet-derived growth factor receptor alpha (Pdgfra) in the movement of myocardial cells have suggested a paracrine chemotaxis model, in which the myocardium senses chemokine signals from the endoderm and responds to them (*Bloomekatz et al., 2017*). However, the existence and identity of these intrinsic myocardial mechanisms remain to be fully elucidated.

We have sought to identify the intracellular pathways downstream of Pdgfra that regulate the collective movement of the myocardium. The phosphoinositide 3-kinase (PI3K) pathway is known as an intracellular signaling mediator of receptor tyrosine kinases (RTKs; e.g. Pdgfra). During PI3K signaling, PI3K phosphorylates phosphatidylinositol (4,5)-bisphosphate (PIP2) to create phosphatidylinositol (3,4,5)-trisphosphate (PIP3), which recruits pleckstrin homology (PH)-domain containing proteins and regulates many cellular processes including proliferation and cell migration (*Fruman et al., 2017*). Both individualistic cell migration such as in *Dictyostelium* and neutrophils (*Iijima and Devreotes, 2002*; *Yoo et al., 2010*) and collective cell migration such as in the movement of border cells in *Drosophila* and the movement of the anterior visceral endoderm during mouse gastrulation (*Ghiglione et al., 2018*; *Bloomekatz et al., 2012*) have been shown to be regulated by PI3K signaling.

Using the advantages of external development and ease of live imaging in the zebrafish model system (*Shrestha et al., 2020*), our studies reveal that myocardial PI3K signaling is required for proper directional movement toward the midline during cardiac fusion. In particular, we find that inhibition of the PI3K pathway, throughout the embryo or only in the myocardium, results in bilateral cardiomyocyte populations that fail to reach the midline (cardia bifida) or have only partially merged by the time wild-type myocardial cells are fully merged. High-resolution live imaging in combination with mosaic labeling further reveals that the orientation of myocardial membrane protrusions during cardiac fusion is dependent on PI3K signaling. Furthermore, we find that PI3K and Pdgfra interact to facilitate cardiac fusion. Altogether our work supports a model by which intrinsic Pdgfra–PI3K signaling regulates the formation of membrane protrusions, facilitating the collective movement of the myocardium toward the midline. Insight into the balance of extrinsic and intrinsic influences for directing collective movement of myocardial cells has implications for understanding a wide set of congenital and environmental cardiac defects as well as the pathogenic mechanisms of diseases broadly associated with collective movement.

## Results

### The PI3K pathway is required for proper cardiac fusion

In a search for intracellular signaling pathways that are important for cardiac fusion, we examined the PI3K signaling pathway by pharmacological inhibition of PI3K activity with LY294002 (LY) (*Vlahos et al., 1994*). Treatments were started at bud stage (10 hours post-fertilization – hpf), in order to exclude effects on mesodermal cells during gastrulation (*Montero et al., 2003*). In wild-type or dimethyl sulfoxide (DMSO)-treated embryos, bilateral myocardial populations move toward the midline and merge to form a ring structure between 20 and 21 hpf, which corresponds to the 20–22 somite stage (s) (*Figure 1A, A', F, G*). However, in embryos treated with 15–25 µM LY myocardial movement is disrupted and the bilateral myocardial populations fail to properly merge by 22s (*Figure 1B, B', F, G*, *Figure 1—figure supplement 1A–C, M*). To ensure our analysis of cardiac fusion phenotypes was not complicated by a developmental delay, we used developmentally stage-matched embryos (somite stage) rather than time-matched embryos (hpf) (see *Figure 1—figure supplement 2* for an analysis using time-matched embryos).

LY targets class I PI3K complexes, which is useful since class I complexes are known to be functionally redundant (*Jean and Kiger, 2014*; *Juss et al., 2012*). However, at higher concentrations than required for inhibiting class I PI3K complexes, LY has also been shown to inhibit non-PI3K complexes (off-targets) (*Gharbi et al., 2007*). To exclude the possibility that LY-induced cardiac fusion phenotypes result from off-target artifacts, we exposed bud stage embryos to two other PI3K inhibitors, Dactolisib (Dac) or Pictilisib (Pic) (*Raynaud et al., 2009*; *Folkes et al., 2008*; *Maira et al., 2008*). Exposure to these PI3K inhibitors (20–50 µM) causes cardiac fusion defects (*Figure 1A–D, A'–D', F, G*, *Figure 1—figure supplement 1D–I, N, O*) and corresponding reductions in PI3K activity, as measured by the ratio of phosphorylated AKT (pAKT) to AKT (*Figure 1H*, *Figure 1—figure supplement 1Q–R*). AKT is phosphorylated as a direct consequence of PI3K activity (*Alessi et al., 1996*). Despite the increased specificity of Dac and Pic, all three inhibitors (Dac, Pic, and LY) have been shown to also inhibit mTOR (*Folkes et al., 2008*; *Maira et al., 2008*; *Brunn et al., 1996*), as well as class I PI3K complexes. Thus, we directly examined a role for mTOR in cardiac fusion. Incubating embryos from bud stage to 22s with rapamycin, an inhibitor of mTOR (*Heitman et al., 1991*), at multiple concentrations did not affect cardiac fusion (*Figure 1—figure supplement 3A–F*), even though mTOR activity as measured by S6 phosphorylation (*Holz and Blenis, 2005*) was dramatically reduced (*Figure 1—figure supplement 3G*). Furthermore, using a non-pharmacological approach we found that mRNA injection of a truncated form of p85, which acts as a dominant negative inhibitor of PI3K (dnPI3K) activity (*Carballada et al., 2001*) also caused cardiac fusion defects (*Figure 1E, E', F, G*, *Figure 1—figure supplement 1J–L, P, S*). Thus, inhibition of PI3K activity with LY, Dac, Pic, and dominant-negative p85 mRNA, but not mTOR causes cardiac fusion defects.

We next characterized the morphology of the cardiac ring in PI3K-inhibited embryos and the cellular processes known to be regulated by PI3K signaling. During the later stages of cardiac fusion, myocardial cells develop epithelial polarity in which proteins such as ZO1 form intercellular junctions between myocardial cells (*Trinh and Stainier, 2004*; *Jackson et al., 2017*). During this time, the myocardium also moves deeper into the tissue eventually residing ventral to the endoderm in a process known as subduction (*Ye et al., 2015*) and it forms a contiguous second dorsal layer (*Figure 1—figure supplement 4A–C*, arrows). In PI3K-inhibited embryos, we found that myocardial cells form this second dorsal layer however, the localization of polarity markers and the tissue organization can appear mildly disorganized (*Figure 1—figure supplement 4D–F*). The PI3K signaling pathway is known to promote cell proliferation and cell survival (*Fruman et al., 2017*) however, we did not find a difference in the number of cardiomyocytes (*Figure 1—figure supplement 4G–I*) nor in the number of cardiomyocytes in S-phase as measured by EdU incorporation (*Figure 1—figure supplement 4J–L*) in PI3K-inhibited embryos at 20s compared to DMSO-treated embryos. Similarly, no apoptotic cardiomyocytes were observed in DMSO- nor in 20 µM LY-treated embryos (*Figure 1—figure supplement 4M–P*, *n* = 17, 19 embryos, respectively, from three biological replicates), despite apoptosis being observed in DNAse-treated controls (*Figure 1—figure supplement 4O, P*).

We further investigated whether increasing PI3K signaling affects cardiac fusion, by examining *ptena, ptenb* mutants. Pten, a lipid phosphatase, opposes PI3K function by converting PIP3 to PIP2 (*Fruman et al., 2017*). Homozygous *ptena*$^{-/-}$ mutants, along with double homozygous *ptena*$^{-/-}$, *ptenb*$^{-/-}$ mutants did not show defects in cardiac fusion (*Figure 1—figure supplement 5A–C, J*).

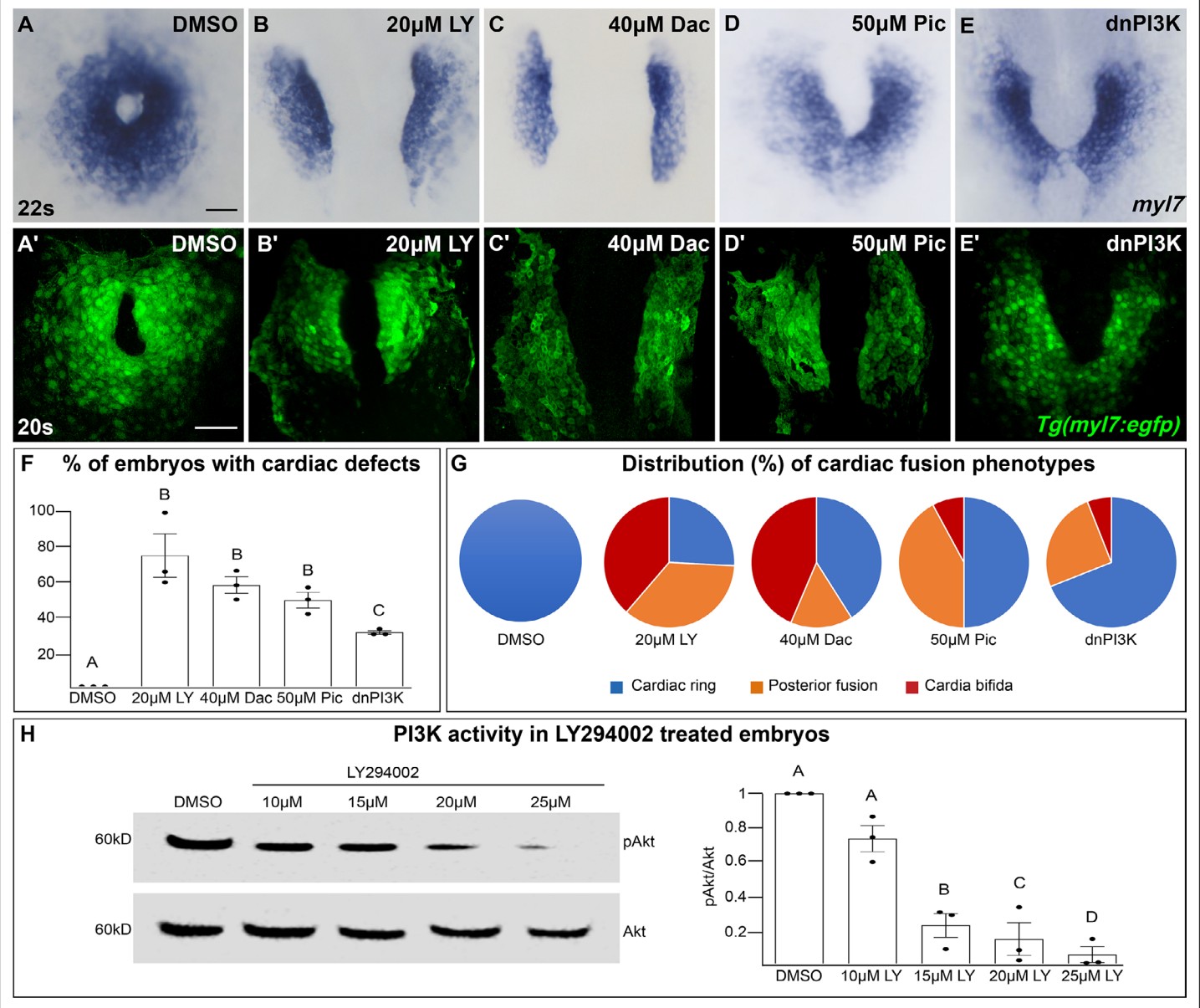

**Figure 1.** The phosphoinositide 3-kinase (PI3K) pathway is required for cardiac fusion. Dorsal views, anterior to the top, of the myocardium labeled with *myl7* (**A–E**) at 22 somite stage (s) or *Tg(myl7:egfp)* (**A'–E'**) at 20s. In contrast to a ring of myocardial cells in DMSO-treated embryos (**A, A'**), in embryos treated with PI3K inhibitors LY294002 (LY, **B, B'**), Dactolisib (Dac, **C, C'**), or Pictilisib (Pic, **D, D'**) at bud stage or injected with *dnPI3K* mRNA (750 pg) at the one-cell stage (**E, E'**) cardiac fusion fails to occur properly with embryos displaying either cardia bifida (**B, C**) or fusion only at the posterior end (**D, E**). Graphs depict the percentage (**F**) and range (**G**) of cardiac fusion defects in control and PI3K-inhibited embryos. Dots represent the percent of embryos with cardiac defects per biological replicate. Total embryos analyzed *n* = 37 (DMSO), 31 (20 μM LY), 39 (40 μM Dac), 38 (50 μM Pic), and 86 (*dnPI3K*). Blue – cardiac ring/normal; orange – fusion only at posterior end/mild phenotype, red – cardia bifida/severe phenotype. (**H**) Representative immunoblot and ratiometric analysis of phosphorylated Akt (pAkt) to Akt protein levels in DMSO- and LY-treated embryos reveals a dose-dependent decrease in PI3K activation. Bar graphs indicate mean ± standard error of the mean (SEM), dots indicate pAKT/AKT ratio per biological replicate, normalized to DMSO. Three biological replicates per treatment. One-way analysis of variance (ANOVA) tests – letter changes indicate differences of p < 0.05 (**F, H**). Scale bars, 40 μm (**A–E**), 42 μm (**A'–E'**). Raw data and full p-values included in the source file.

The online version of this article includes the following source data and figure supplement(s) for figure 1:

**Source data 1.** Statistical source data for *Figure 1F, H*.

**Source data 2.** Original immunoblots used in *Figure 1H* (raw, uncropped) with and without labeling.

**Figure supplement 1.** The penetrance and severity of cardiac fusion defects in phosphoinositide 3-kinase (PI3K)-inhibited embryos is dose dependent.

**Figure supplement 1—source data 1.** Statistical source data for *Figure 1—figure supplement 1M–O, P–S*.

*Figure 1 continued on next page*

*Figure 1 continued*

**Figure supplement 1—source data 2.** Original immunoblots used in *Figure 1—figure supplement 1Q–S* (raw, uncropped) with and without labeling.

**Figure supplement 2.** LY incubation results in trunk extension and somite formation delays.

**Figure supplement 2—source data 1.** Statistical source data for *Figure 1—figure supplement 2C, F, I*.

**Figure supplement 3.** Inhibition of mTOR activity does not affect cardiac fusion.

**Figure supplement 3—source data 1.** Statistical source data for *Figure 1—figure supplement 3F, G*.

**Figure supplement 3—source data 2.** Original immunoblots used in *Figure 1—figure supplement 3G* (raw, uncropped) with and without labeling.

**Figure supplement 4.** Morphology and proliferation in the myocardium are not compromised in phosphoinositide 3-kinase (PI3K)-inhibited embryos.

**Figure supplement 4—source data 1.** Statistical source data for *Figure 1—figure supplement 4I, L, M–P*.

**Figure supplement 5.** Loss of Pten, an antagonist of phosphoinositide 3-kinase (PI3K) activity, causes cardiac fusion defects.

**Figure supplement 5—source data 1.** Statistical source data for *Figure 1—figure supplement 5J*.

**Figure supplement 6.** Inhibition of Pten activity with VO-OHpic increases pAkt and causes cardiac fusion defects.

**Figure supplement 6—source data 1.** Statistical source data for *Figure 1—figure supplement 6F–H*.

**Figure supplement 6—source data 2.** Original immunoblots used in *Figure 1—figure supplement 6H* (raw, uncropped) with and without labeling.

However, maternal contribution of *Pten* has been reported to persist till at least 60 hpf (*Choorapoikayil et al., 2013*; *Faucherre et al., 2008*), possibly explaining the lack of phenotype in these mutants. Indeed, cardiac fusion defects have been observed in *pten* mutant mouse embryos (*Bloomekatz et al., 2012*) and we found that inhibiting Pten in zebrafish embryos from bud stage to 22s with the small molecule Pten inhibitor VO-OHpic (*Rosivatz et al., 2006*) does result in cardiac fusion defects, along with increased PI3K activity (*Figure 1—figure supplement 6*). Furthermore, adding a sub-phenotypic dose of VO-OHpic (*Figure 1—figure supplement 5D–F*) to double homozygous *ptena*$^{-/-}$, *ptenb*$^{-/-}$ mutant embryos to address the role of the remaining maternal contribution also results in cardiac fusion defects (*Figure 1—figure supplement 5G–J*, Fisher's test p-value 8.21E−06). These findings reveal that loss-of-Pten which causes increased PIP3 levels results in cardiac fusion defects and together with our PI3K inhibition experiments indicate that appropriate levels of PI3K signaling are required for proper cardiac fusion.

## The extent and duration of PI3K inhibition determine the penetrance and severity of cardiac fusion defects

PI3K-inhibited embryos display cardiac phenotypes at 22s that range from severe, in which the myocardial populations remain entirely separate (cardia bifida) (*Figure 1G* – red; examples – *Figure 1—figure supplement 1C, F, I, L*), to more mildly affected hearts in which the myocardial populations form a U-shaped structure, having merged at the posterior but not anterior end (*Figure 1G* – orange; examples *Figure 1—figure supplement 1B, E, H, K*). A subset of the PI3K-inhibited embryos also appear phenotypically normal (~25% for 20 µM LY, *Figure 1F, G*) indicating incomplete penetrance. However, increasing the concentration of PI3K inhibitor or dnPI3K mRNA increases the severity and penetrance of these phenotypes in a dose-dependent manner (*Figure 1—figure supplement 1*). Similarly, we confirmed that LY inhibits PI3K activity in a dose-dependent manner (*Figure 1H*). Thus, the severity and penetrance of cardiac fusion defects depend on the efficacy of PI3K inhibition.

Since differing modes of movement (*Holtzman et al., 2007*) and cellular processes such as MET (*Trinh and Stainier, 2004*; *Jackson et al., 2017*) and subduction (*Ye et al., 2015*) occur at distinct developmental stages during cardiac fusion, we evaluated the developmental stages over which PI3K signaling is required. Short exposures (<3 hr) just prior to 22s or starting at bud stage had no effect on cardiac fusion. However, progressively longer times of exposure ending at 22s or starting at bud stage result in correspondingly more severe phenotypes and higher penetrance (*Figure 2A, B*). These addition and wash-out experiments indicate that both the severity and penetrance of cardiac fusion phenotypes correlate with the duration of LY incubation and not a specific developmental stage inside the 3–20s window. Thus, the translocation of the myocardium toward the midline is responsive to both the levels and duration of PI3K signaling throughout cardiac fusion.

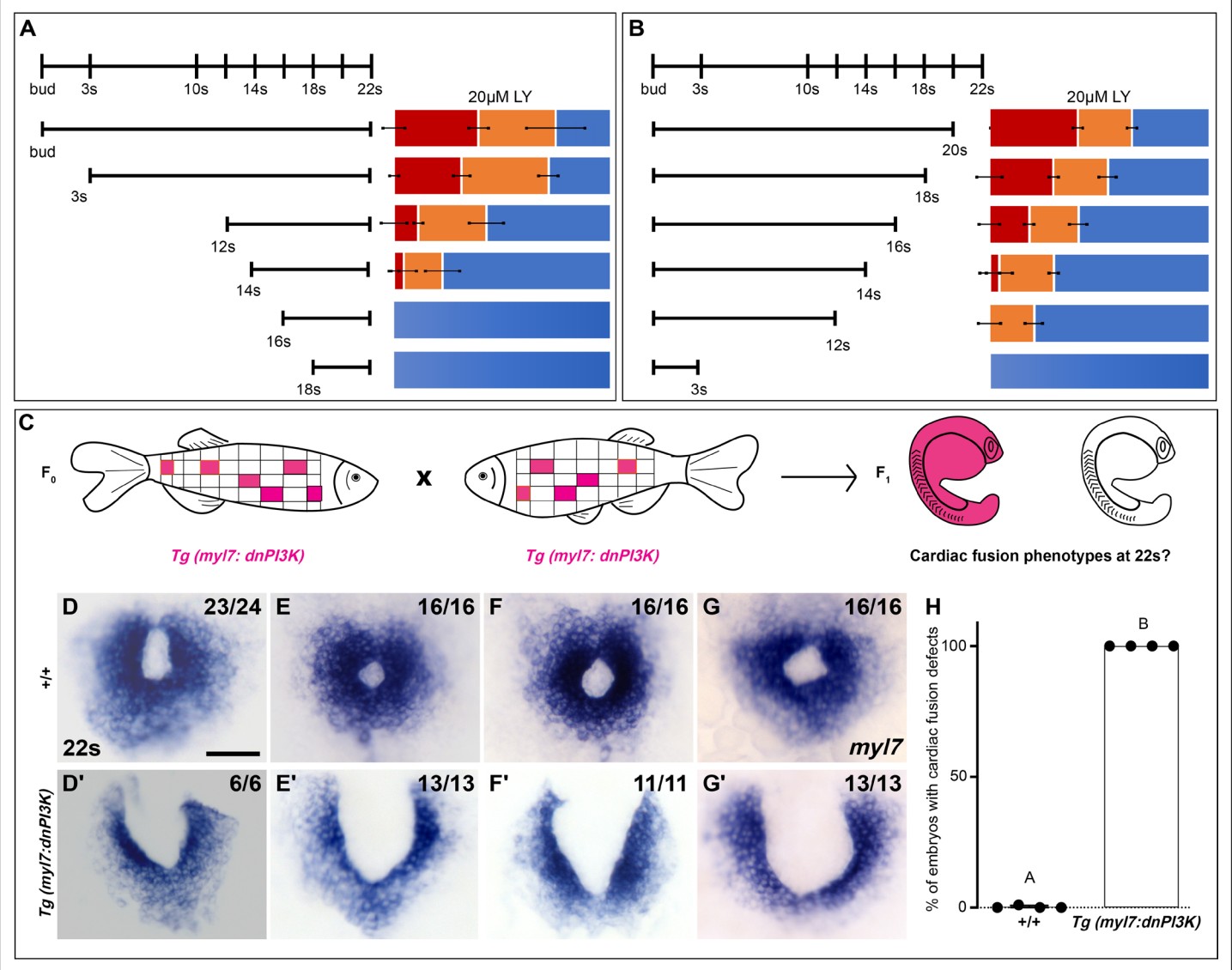

**Figure 2.** Phosphoinositide 3-kinase (PI3K) is required in the myocardium throughout cardiac fusion. Graphical representation of the PI3K inhibitor addition (**A**) and wash-out (**B**) experiments used to determine the developmental stage over which PI3K is required. In (**A**) LY is added to embryos at different developmental stages and incubated until 22s, when cardiac fusion is assessed. In (**B**), LY is added at bud stage and washed-out at different developmental stages, after which embryos are incubated in normal media till 22s, when cardiac fusion is assessed. Bar graphs indicate the average proportion of embryos displaying different phenotypes. Blue – cardiac ring/normal; orange – fusion only at posterior end/mild phenotype, red – cardia bifida/severe phenotype. n = 45 embryos per treatment condition from three biological replicates. (**C**) Schematic outlines experimental design to test requirement for PI3K in the myocardium. Pink – cells with the *Tg(myl7:dnPI3K)* transgene. F0 animals are mosaic for the transgene, while all cells in F1 embryos either have the transgene (pink) or do not (white). The *myl7* promoter restricts *dnPI3K* expression to the myocardium in *Tg(myl7:dnPI3K)* embryos. (**D–G**) Dorsal view of the myocardium labeled with *myl7* in embryos at 22s from four different founder pairs (**D–D', E–E', F–F', G–G'**). F1 embryos without the *Tg(myl7:dnPI3K)* transgene (as determined by genotyping) display normal cardiac fusion (D–G, n = 23/24, 16/16, 16/16, 16/16, per founder pair), while F1 siblings with the *Tg(myl7:dnPI3K)* transgene display cardiac fusion defects (D'–G', n = 6/6, 13/13, 11/11, 13/13), indicating that PI3K signaling is required in myocardial cells. (**H**) Graph indicating the average % of wild-type and *Tg(myl7:dnPI3K)+* embryos with cardiac fusion defects. Letter difference indicates a significant Fisher's exact test, p = 5.56 × 10⁻³¹. Scale bar, 40 μm.

The online version of this article includes the following video, source data, and figure supplement(s) for figure 2:

**Source data 1.** Statistical source data for *Figure 2*.

**Figure supplement 1.** The morphology of endoderm is not compromised in phosphoinositide 3-kinase (PI3K)-inhibited embryos.

**Figure supplement 1—source data 1.** Statistical source data for *Figure 2—figure supplement 1D, H*.

**Figure supplement 2.** Phosphoinositide 3-kinase (PI3K) activity in myocardial cells.

*Figure 2 continued on next page*

*Figure 2 continued*

**Figure supplement 2—source data 1.** Statistical source data for *Figure 2—figure supplement 2C–E*.

**Figure 2—video 1.** PH-mkate2 is localized asymmetrically at the membrane of myocardial cells in DMSO-treated embryos, but is found in the cytoplasm and subcellular organelles in LY-treated embryos.

https://elifesciences.org/articles/85930/figures#fig2video1

## PI3K signaling is required in the myocardium for proper cardiac fusion

Mutations affecting the specification or morphology of the anterior endoderm result in myocardial movement defects (*Kupperman et al., 2000*; *Ye and Lin, 2013*; *Osborne et al., 2008*; *Fukui et al., 2014*), revealing a non-autonomous role for the anterior endoderm in cardiac fusion. However, when PI3K signaling is inhibited with 15 or 25 µM LY starting at bud stage we did not observe differences in the expression of endoderm markers such as *axial/foxa2* or *Tg(sox17:egfp)* compared to DMSO-treated embryos (*Figure 2—figure supplement 1A–C, E–G*). A detailed examination of the anterior endoderm morphology in PI3K-inhibited embryos revealed it was intact and contiguous similar to DMSO-treated embryos (*Figure 2—figure supplement 1I, J*) as was the average anterior endoderm width (*Figure 2—figure supplement 1D, H*). Thus, cardiac fusion defects in PI3K-inhibited embryos are unlikely to be due to changes in the anterior endoderm.

To determine if PI3K signaling is specifically required within the myocardium, as opposed to the endoderm, we created a myocardial-specific dominant negative transgenic construct, *Tg(myl7:dnPI3K)*. Our experimental design is outlined in *Figure 2C*. In F1 embryos at 22s, derived from incrosses between *Tg(myl7:dnPI3K)* mosaic F0 animals, we observed embryos with normal cardiac rings and embryos with cardiac fusion defects (*Figure 2D–G'*). Genotyping revealed that F1 embryos with normal cardiac rings (*Figure 2D–G*) did not have the transgene ($n$ = 71/71), while almost all sibling embryos with cardiac fusion defects (*Figure 2D'–G'*) were positive for the *Tg(myl7:dnPI3K)* transgene ($n$ = 43/44). And all embryos with the *Tg(myl7:dnPI3K)* transgene have a cardiac fusion defect (*Figure 2H*). (F1 embryos from four independent founder pairs were analyzed. Stable transgenics could not be propagated due to loss of viability, likely due to a requirement for PI3K signaling in cardiac contraction at later stages; *Crackower et al., 2002*.) Statistical analysis reveals that the *Tg(myl7:dnPI3K)* transgene is significantly associated with a cardiac fusion defect (Fisher's test $p = 5.56 \times 10^{-31}$).

We further confirmed that PI3K signaling is active in the myocardium during cardiac fusion by using a PH domain reporter (*Balla and Várnai, 2009*). Fluorophores fused to PH-domains from PIP3-binding proteins translocate to the membrane when PIP3 is induced. For example, the PH-domain from BTK fused to mkate2 localizes to the region of the plasma membrane where PI3K is actively creating PIP3 (*Hall et al., 2020*). By expressing this reporter (*myl7:PH-mkate2*) mosaically in the myocardium, we found that PH-mkate2 was enriched asymmetrically at intercellular plasma membrane boundaries in DMSO-treated embryos at 20s (*Figure 2—figure supplement 2A–A'''*, *C–E*, *Figure 2—video 1A*). However, in PI3K-inhibited embryos PH-mkate2 was localized diffusely throughout the cytoplasm or in subcellular compartments (*Figure 2—figure supplement 2B–E*, *Figure 2—video 1B*). Combined with our tissue-specific inhibition of PI3K signaling these findings suggest that PI3K signaling acts within the myocardium to regulate its movement during cardiac fusion.

## PI3K signaling is responsible for the steering and velocity of myocardial movements during cardiac fusion

Our analysis points to a role for PI3K signaling in the movement of myocardial cells. To identify the properties of myocardial movement regulated by PI3K signaling, we analyzed myocardial movement by performing in vivo time-lapse imaging with the *Tg(myl7:egfp)* transgene, which labels myocardial cells. A time-series using *hand2* expression reveals dramatic differences between DMSO- and LY-treated embryos in the translocation of the myocardium beginning after 12s (*Figure 3—figure supplement 1*). We thus focused our time-lapse imaging on the 14–20s developmental window. In time-lapse movies of DMSO-treated embryos, myocardial cells display coherent medially directed movement (*Figure 3A–B, E*, *Figure 3—figure supplement 2A–A'''*, *Figure 3—video 1*) with an average velocity of 0.2334 ± 0.007 µm/min, which is consistent with previous studies (*Holtzman et al., 2007*; *Bloomekatz et al., 2017*). In PI3K-inhibited embryos myocardial cells also display coherent, coordinated movement and do move in the general direction of the midline, however they make

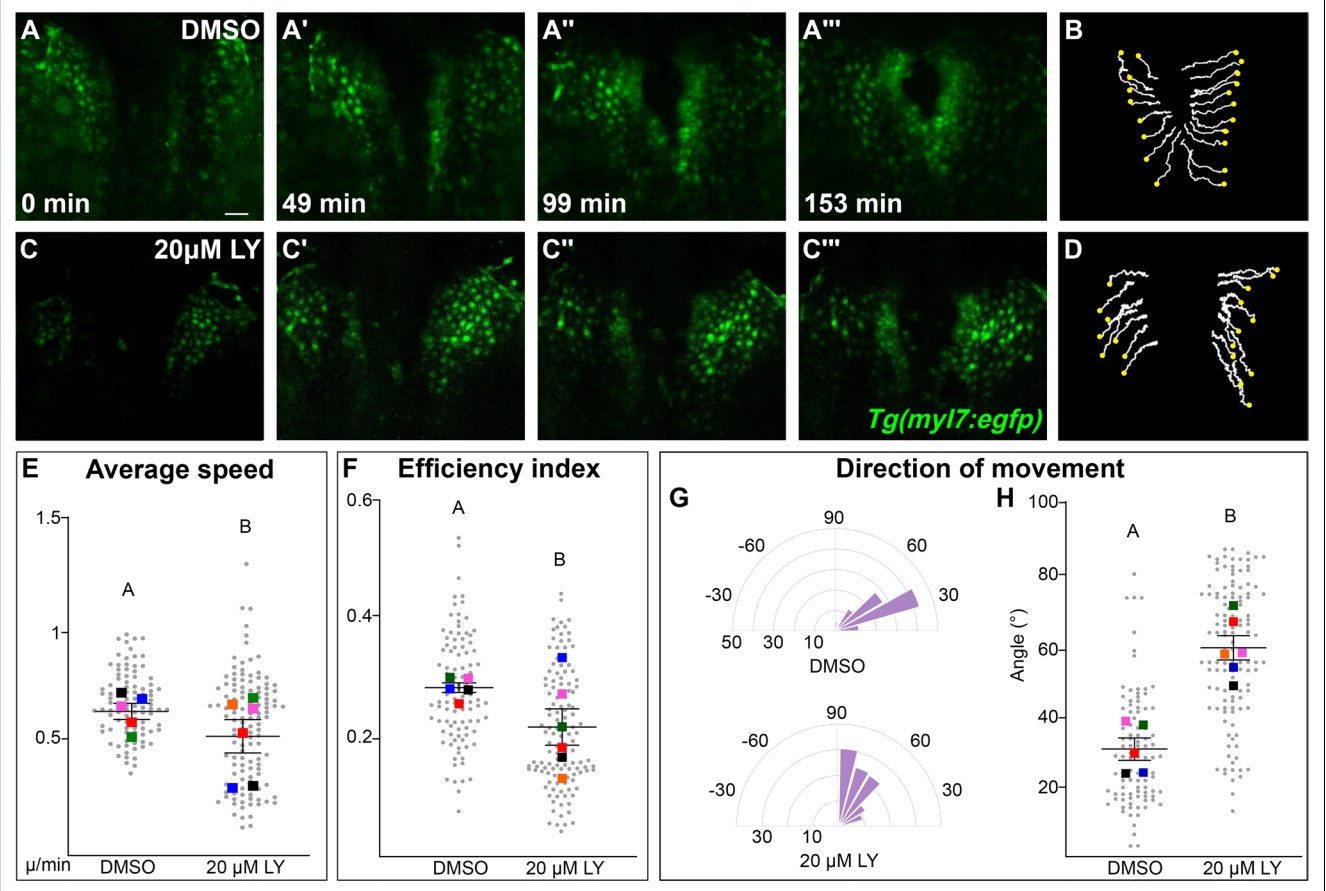

**Figure 3.** Phosphoinositide 3-kinase (PI3K) signaling regulates the medial movement and speed of the myocardium during cardiac fusion. Time points from a representative video of myocardial cells visualized with the *Tg(myl7:egfp)* transgene in embryos treated with DMSO (**A, B**, *Figure 3—video 1*) or 20 μM LY (**C, D**, *Figure 3—video 2*) from bud stage to 22s. 3D reconstructions of confocal slices (**A, C**) reveal changes in conformation and location of the myocardium at three major stages of cardiac fusion: early medial movement toward the embryonic midline (**A–A', C–C'**), posterior merging of bilateral populations (**A'', C''**) and anterior merging to form a ring (**A''', C'''**). Representative tracks (**B, D**) show the paths of a subset of myocardial cells over ~2.5 hr. Yellow dots indicate the starting point of each track. Graphs depict average speed (**E**), efficiency index (**F**), and angle of movement (**G, H**) of myocardial cells. Angular movement along the anterior–posterior axis does not distinguish anterior from posterior movement (**G, H**). Myocardial cells in PI3K-inhibited embryos show an overall direction of movement that is angular (60–90°) and is slower than in DMSO-treated embryos. 96 and 125 cells were analyzed from five DMSO- and six 20 μM LY-treated embryos, respectively. Gray dots – individual cells; color squares – average per embryo. Average of embryos and standard error (shown in **E, F, H**). Two-sample *t*-test, letter change indicates p < 0.05. Scale bars, 60 μm. Quantification details in the methods. Raw data and full p-values included in the source file.

The online version of this article includes the following video, source data, and figure supplement(s) for figure 3:

**Source data 1.** Statistical source data for quantification of myocardial movement behaviors in *Figure 3E–H*.

**Figure supplement 1.** Myocardial movement toward the midline is disrupted in phosphoinositide 3-kinase (PI3K)-inhibited embryos throughout cardiac fusion.

**Figure supplement 1—source data 1.** Statistical source data for distance between bilateral anterior lateral plate mesoderm (ALPM) populations, *Figure 3—figure supplement 1I*.

**Figure supplement 2.** Phosphoinositide 3-kinase (PI3K) signaling directs myocardial movement during the early stages of cardiac fusion and regulates velocity along the medial-lateral axis.

**Figure supplement 2—source data 1.** Statistical source data for quantification of myocardial movement separated by stages of movement and location, *Figure 3—figure supplement 2C–F*.

**Figure 3—video 1.** Myocardial cells in DMSO-treated embryos collectively move toward the midline and form a ring during cardiac fusion.
https://elifesciences.org/articles/85930/figures#fig3video1

**Figure 3—video 2.** Myocardial cells in phosphoinositide 3-kinase (PI3K)-inhibited embryos fail to move properly toward the midline.
https://elifesciences.org/articles/85930/figures#fig3video2

*Figure 3 continued on next page*

*Figure 3 continued*

**Figure 3—video 3.** Phosphoinositide 3-kinase (PI3K) signaling promotes the medial directional movement of myocardial cells toward the midline.
https://elifesciences.org/articles/85930/figures#fig3video3

dramatically less progress (*Figure 3C, D*, *Figure 3—figure supplement 2B–B'''*, *Figure 3—video 2*). Quantitative analysis of these myocardial cell tracks reveals that myocardial cells are slower ($0.1879 \pm 0.008$ μm/min) and less efficient (*Figure 3E, F*, *Figure 3—video 3*). A close analysis of the differences in speed found that they occur throughout cardiac fusion (*Figure 3—figure supplement 2C*) and are mostly due to defective movement along the medial–lateral axis (*Figure 3—figure supplement 2E*) rather than defects in angular movement occurring along the anterior–posterior axis (*Figure 3—figure supplement 2F*).

The most dramatic difference between PI3K-inhibited and DMSO-treated myocardial cells is in the direction of their movement. Tracks of myocardial cells in DMSO-treated embryos are predominately oriented in a medial direction (average of $31.1 \pm 1.65°$), while tracks in LY-treated embryos are mostly oriented in an angular anterior direction ($60.6 \pm 1.73°$, p-value = $2.77 \times 10^{-12}$, *Figure 3G, H*). These differences occur mainly in the early stages of cardiac fusion when wild-type myocardial movement is mostly medial (*Figure 3—figure supplement 2D*). Altogether, this analysis of myocardial cell tracks suggests that PI3K signaling is responsible for both steering and propelling myocardial cells toward the midline.

## Myocardial membrane protrusions are medially polarized by PI3K signaling

The role of PI3K signaling in regulating the polarity of migratory protrusions in the dorsal epithelium of *Drosophila* embryos and in the prechordal plate of zebrafish embryos (*Montero et al., 2003*; *Garlena et al., 2015*) along with previous reports of the existence of myocardial membrane protrusions during heart tube formation (*Dominguez et al., 2023*; *Ye et al., 2015*) led us to look for these protrusions during cardiac fusion and to examine if they are disrupted in PI3K-inhibited embryos. To visualize membrane protrusions in the myocardium, we performed in vivo time-lapse imaging during cardiac fusion of embryos injected with *myl7:lck-egfp* plasmid DNA in order to mosaically label the plasma membrane of myocardial cells. Despite myocardial cells being connected via intercellular junctions (*Trinh and Stainier, 2004*; *Bloomekatz et al., 2017*), we observed that the lateral edges of myocardial cells in wild-type/DMSO-treated embryos are highly dynamic; transitioning from appearing smooth and coherent to undulating and extending finger-like membrane protrusions away from the cell (*Figure 4A–A''''*, *Figure 4—video 1*). These protrusions actively extend and retract, and are prevalent occurring on average $20.3 \pm 6.7$ times per hour per cell and lasting for an average of $2.3 \pm 0.6$ min (*Figure 4A*). We observed different types of protrusion morphologies, including thin finger-like protrusions and wide protrusions which have a wider base (*Figure 4—figure supplement 1A*). In LY-treated embryos we observed similar types of membrane protrusions extending from myocardial cells (*Figure 4B–B''''*, *Figure 4—figure supplement 1B*, *Figure 4—video 1*). These protrusions occur at a similar rate to those found in DMSO-treated embryos ($17 \pm 7.4$ per hour per cell, p-value = 0.36), but with slightly longer persistence ($3.23 \pm 0.84$ min, p-value = 0.008).

We further observed that both types of membrane protrusions in DMSO-treated embryos occur predominantly in the medial direction ($77.25 \pm 21.76\%$ of protrusions were in the forward/medial direction, *Figure 4A–A''''*, *C, D*, and *Figure 4—figure supplement 1C, D*), suggesting an association with the medial movement of the myocardial tissue. In contrast, in LY-treated embryos myocardial membrane protrusions do not display the same medial polarity, instead extending from all sides of a myocardial cell equally (only $46 \pm 11.6\%$ of protrusion were in the forward direction, *Figure 4B–B''''*, *C, D* and *Figure 4—figure supplement 1C, D*). The finding that myocardial membrane protrusions are medially polarized in wild-type embryos but not in PI3K-inhibited embryos where myocardial cells are misdirected and slower to reach the midline suggests that PI3K signaling helps to steer and propel myocardial cells toward the midline through the polarization of these active protrusions.

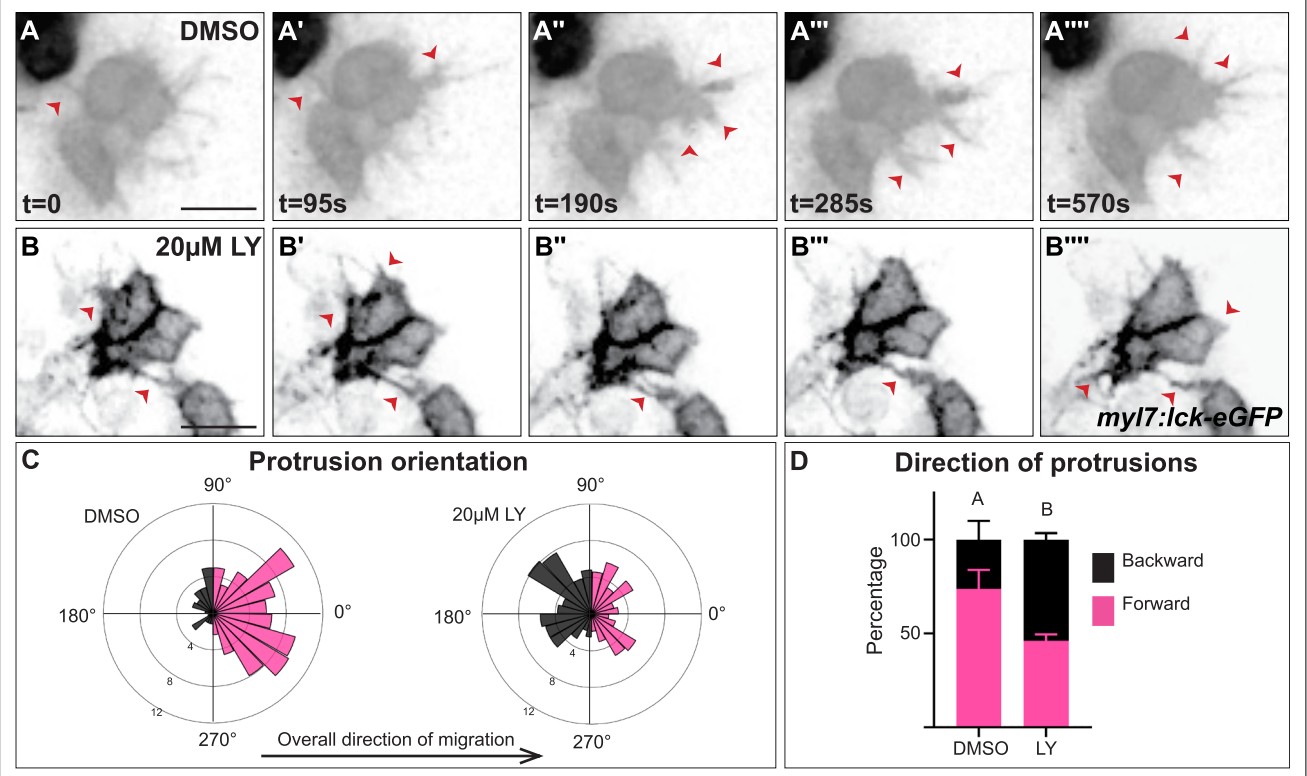

**Figure 4.** Myocardial membrane protrusions are misdirected in phosphoinositide 3-kinase (PI3K)-inhibited embryos. (**A–B''''**) timepoints from representative videos (see *Figure 4—video 1*) of myocardial cells whose membrane has been labeled with *myl7*:lck-eGFP (black), medial to the right, in a DMSO- (**A–A''''**) or a 20 µM LY- (**B–B''''**) treated embryo. Red arrowheads indicate representative protrusions, which are mostly oriented medially, coincident with the direction of movement in DMSO-treated embryos (**A–A''''**) but are oriented in all directions in LY-treated embryos (**B–B''''**). Rose (**C**) and bar (**D**) graphs displaying the orientation of membrane protrusions in DMSO- (left) or LY- (right) treated embryos. The length of each radial bar in (**C**) represents the percentage of protrusions in each bin. Bar graph displays the total percentage of forward or backward protrusions. Forward protrusions: 270–90°, pink. Backward protrusions: 90–270°, black. $n$ = 425 protrusions from 11 cells in 5 embryos (DMSO), and 480 protrusions from 11 cells in 4 embryos (20 µM LY). Fisher's exact test, p-value $1.8 \times 10^{-5}$. Mean ± standard error. Scale bar, 30 µm. Raw data included in the source file.

The online version of this article includes the following video, source data, and figure supplement(s) for figure 4:

**Source data 1.** Statistical source data for quantification of myocardial protrusion properties.

**Figure supplement 1.** Different types of myocardial protrusion morphologies occur during cardiac fusion.

**Figure supplement 1—source data 1.** Statistical and raw source data for *Figure 4—figure supplement 1C, D*.

**Figure 4—video 1.** Dynamic medially oriented myocardial membrane protrusions are lacking in phosphoinositide 3-kinase (PI3K)-inhibited embryos.
https://elifesciences.org/articles/85930/figures#fig4video1

## PI3K signaling is regulated by Pdgfra during cardiac fusion

We next investigated whether Pdgfra activates PI3K signaling to regulate myocardial movement. We found that PI3K activity as measured by the ratio of phospho-AKT to AKT levels (*Alessi et al., 1996*) is severely diminished in *pdgfra* mutant embryos during cardiac fusion (*Figure 5A*). Conversely, when Pdgfra activity is increased during cardiac fusion through the over-expression of *pdgfaa*, PI3K activity is upregulated (*Figure 5B*).

To determine if Pdgfra's influence on PI3K activity is important for myocardial movement toward the midline, we tested whether they functionally interact to regulate cardiac fusion. When *pdgfra* heterozygous mutant embryos are exposed to DMSO cardiac fusion occurs normally (*Figure 5D, F*), even though there is a small reduction in PI3K activity (*Figure 5A*). Similarly, in wild-type embryos exposed to 10 µM LY, PI3K activity is modestly reduced (*Figure 1H*) and a small percent of embryos display mild cardiac fusion defects (average of 10.9 ± 7.39% of 10 µM LY-treated embryos display mild U-shaped cardiac fusion defects, $n$ = 36, 3 replicates, *Figure 5C, F*). However, when *pdgfra* heterozygous mutant embryos are exposed to 10 µM LY, there is a synergistic increase in both the

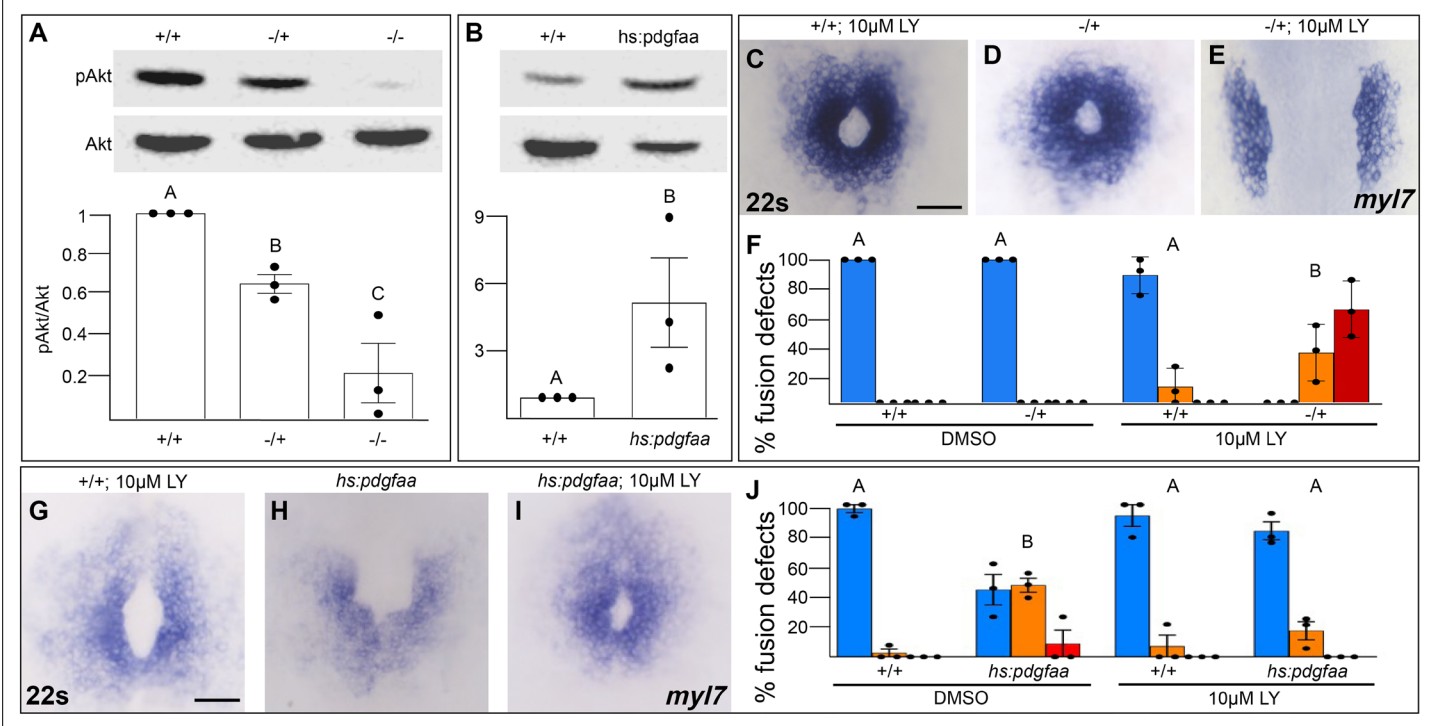

**Figure 5.** Pdgfra activates and genetically interacts with phosphoinositide 3-kinase (PI3K) signaling to regulate cardiac fusion. (**A, B**) Immunoblot and ratiometric analysis of phosphorylated Akt (pAkt) compared to total Akt levels reveals reduced pAkt levels in loss-of-function *pdgfra^sk16* heterozygous (−/+) or homozygous (−/−) mutant embryos at 22s (**A**), and elevated pAkt levels at 22s when PDGF signaling is activated with the *Tg(hs:pdgfaa)* transgene (**B**). Bar graphs display averages from three separate experiments. (**C–E, G–I**) Dorsal views, anterior to the top, of the myocardium labeled with *myl7* at 22s. In contrast to a normal ring of myocardial cells found in wild-type embryos treated with 10 µM LY starting at bud stage (**C**) or in *pdgfra* heterozygous embryos (**D**), when *pdgfra* heterozygous mutants are exposed to 10 µM LY, cardiac fusion is defective with embryos displaying severe phenotypes such as cardia bifida (**E**). Furthermore, the percent of cardiac fusion defects observed in *Tg(hs:pdgfaa)* embryos heat-shocked at bud stage (**H**) is greatly decreased when heat-shocked *Tg(hs:pdgfaa)* embryos are exposed to 10 µM LY at bud stage (**I**). (**F, J**) Bar graphs depict the average distribution of cardiac fusion defects from the indicated genotypes. The total number of embryos examined over three separate replicates are 47 (DMSO, +/+), 25 (DMSO, −/+), 36 (10 µM LY, +/+), 31 (10 µM LY, −/+), 49 (heat-shock, DMSO, +/+), 57 (heat-shock, DMSO, *Tg(hs:pdgfaa)*), 51 (heat-shock, 10 µM LY, +/+), 57 (heat-shock, 10 µM LY, *Tg(hs:pdgfaa)*). Blue – cardiac ring/normal; orange – fusion only at posterior end/mild, red – cardia bifida/severe. Bar graphs, mean ± standard error. One-way analysis of variance (ANOVA) (**A, F, J**) or Student's *t*-test (**B**), letter change indicates $p < 0.05$. Scale bar, 60 µm. Raw data and full p-values included in the source file.

The online version of this article includes the following source data for figure 5:

**Source data 1.** Statistical source data for *Figure 5A, B, F, J*.

**Source data 2.** Original immunoblots used in *Figure 5A, B* (raw, uncropped) with and without labeling.

severity and penetrance of cardiac fusion defects. 100% of *pdgfra* heterozygous embryos exposed to 10 µM LY display cardiac fusion defects, with the majority of embryos displaying severe cardia bifida phenotypes (*Figure 5E, F*). Furthermore, we found that cardiac fusion defects caused by exogenous expression of *pdgfaa* (*Figure 5H, J*, *Bloomekatz et al., 2017*), which causes increased PI3K activity (*Figure 5B*), could be rescued by a sub-phenotypic reduction in PI3K activity (*Figure 5I, J*). Together these results suggest that PDGF signaling activates PI3K activity to promote myocardial movement toward the midline.

## Discussion

Our studies reveal an intrinsic PI3K-dependent mechanism by which the myocardium moves toward the midline during the formation of the primitive heart tube. Together with our previous studies revealing a role for the PDGF pathway in facilitating communication between the endoderm and myocardium (*Bloomekatz et al., 2017*), our current work suggests a model in which Pdgfra in the myocardium senses signals (PDGF ligands) from the endoderm and via the PI3K pathway directs

myocardial movement toward the midline through the production of medially oriented membrane protrusions. While genetic and imaging studies in zebrafish and mice (*Trinh and Stainier, 2004*; *Kupperman et al., 2000*; *Molkentin et al., 1997*; *Ye and Lin, 2013*; *Cui et al., 2009*; *Ye et al., 2015*; *Aleksandrova et al., 2015*; *Garavito-Aguilar et al., 2010*; *Arrington and Yost, 2009*) along with embryological studies in chicks and rats (*Goss, 1935*; *Rosenquist, 1970*; *Varner and Taber, 2012*; *Moreno-Rodriguez et al., 2006*) have identified the importance of extrinsic influences – such as the endoderm and extracellular matrix, on myocardial movement to the midline, our studies using tissue-specific techniques identifies an active role for myocardial cells, providing insight into the balance of intrinsic and extrinsic influences that regulate the collective movement of the myocardial tissue during heart tube formation.

Specifically, we found a requirement for PI3K signaling in cardiac fusion. By examining both loss-of-PI3K activity (*Figure 1*) and gain-of-PI3K activity through loss-of-*pten* (*Figure 1—figure supplements 5 and 6*) our studies indicate that appropriate PIP3 levels are required for the proper movement of cardiomyocytes to the midline. This analysis is complemented by previous studies in mice examining *Pten* mutants (*Bloomekatz et al., 2012*). Our spatial and temporal experiments further reveal a requirement for PI3K activity specifically in the myocardium and throughout the duration of cardiac fusion (*Figure 2*). Furthermore, the lack of phenotype caused by inhibition of mTOR (*Figure 1—figure supplement 3*) suggests that mTOR and its downstream signals are unlikely to be important PI3K effectors in regulating cardiomyocyte movement.

When we examined the cellular behaviors affected by loss of PI3K signaling we observed a mild disorganization of myocardial intercellular junctions (*Figure 1—figure supplement 4A–F*). This finding is consistent with previous studies linking epithelial polarity to PI3K signaling (*Krahn, 2020*). However, myocardial cells defective in apical–basal polarity still form a cardiac ring (*Horne-Badovinac et al., 2001*; *Rohr et al., 2006*), suggesting that an apical–basal defect is unlikely to be the primary reason for myocardial movement defects. Instead, our studies showing that PI3K-inhibited myocardial cells move slower and are misdirected during the early stages of cardiac fusion indicate a role for PI3K signaling in the steering of myocardial movements medially toward the midline. Specifically, we found that DMSO-treated myocardial cells in our study display an average direction of 31.1 ± 1.65° compared to PI3K-inhibited embryos which display an average of 60.6 ± 1.73°. Our wild-type tracks results are grossly similar to previous published studies (*Holtzman et al., 2007*; *Bloomekatz et al., 2017*). Differences in the extent of the medial orientation between these studies are likely due to experimental variations in normalization used to account for drift as well as small differences in the developmental stages over which cells were tracked. Our finding that steering in PI3K-inhibited embryos is perturbed in the early stages of cardiac fusion is consistent with the previous identification of different types of myocardial movement during different stages of cardiac fusion (*Holtzman et al., 2007*) and suggests that PI3K signaling could be part of a distinct molecular mechanism that drives these early medial phases of myocardial movement. Loss-of-function *pdgfra* mutants, like PI3K-inhibited embryos, also display defects in the directional movement of myocardial cells (*Bloomekatz et al., 2017*), although the phenotype in *pdgfra* mutants can be more severe. These differences in severity could be a result of differences in the extent of loss-of-function and/or differences in the ability of other genes to compensate. Indeed, our interaction studies (*Figure 5*) suggest that Pdgfra and PI3K signaling work together to regulate cardiac fusion.

Myocardial membrane protrusions during cardiac fusion were postulated by De Haan et al. in 1967 as a mechanism by which myocardial cells move toward the midline (*DeHaan, 1967*). Using mosaic membrane labeling of myocardial cells to visualize membrane protrusions, we have observed myocardial membrane protrusions that are oriented in the medial direction in a PI3K-dependent manner, confirming De Haan's hypothesis. These studies are complemented by previous studies in zebrafish which observed myocardial protrusions prior to and after cardiac fusion (*Ye et al., 2015*; *Rohr et al., 2008*) as well as recent studies in the mice (*Dominguez et al., 2023*) indicating that these cellular processes are likely conserved. Myocardial membrane protrusions have been discovered during trabeculation (*Staudt et al., 2014*), cardiac regeneration (*Morikawa et al., 2015*; *Aharonov et al., 2020*) and during the muscularization of the outflow tract (*van den Hoff et al., 1999*). Indeed, similar observations of PI3K signaling orienting and stimulating protrusion formation in migrating

*Dictyostelium* and neutrophil cells as well as in the collective movement of endothelial tip cells, the prechordal plate, and the dorsal epithelium (*Iijima and Devreotes, 2002*; *Yoo et al., 2010*; *Montero et al., 2003*; *Garlena et al., 2015*; *Dumortier et al., 2012*; *Graupera et al., 2008*) support a conserved role for PI3K signaling in regulating protrusion formation.

However, the question of how active membrane protrusions facilitate the collective medial movement of the myocardium to the midline remains to be addressed. Our studies indicate that directionality and to a lesser extent velocity and efficiency are compromised, when membrane protrusions are improperly oriented in PI3K-inhibited embryos (*Figure 3*). These observations could suggest that the observed membrane protrusions are force generating, similar to protrusions from leader cells in the lateral line or in endothelial and tracheal tip cells (*Caussinus et al., 2008*; *Dalle Nogare et al., 2020*; *Qin et al., 2021*). Alternatively, these protrusions could act more like filopodia sensing extrinsic signals and the extracellular environment (*Heckman and Plummer, 2013*). Future studies examining myocardial protrusions and their role in the biomechanical dynamics of the myocardium will help to elucidate the role of membrane protrusions in the collective movement of the myocardium during cardiac fusion.

Our studies utilize small pharmacological molecules to investigate the role of PI3K signaling in cardiac fusion (*Figure 1*). These inhibitors have been widely used and have several advantages including the ability to target multiple class I PI3K complexes which are known to compensate for each other (*Juss et al., 2012*), and the ability to avoid earlier pleiotropic effects by temporally restricting their usage (*Montero et al., 2003*; *Leslie et al., 2007*). Nevertheless, small molecule inhibitors can also affect non-targeted proteins resulting in off-target phenotypic artifacts. We have tried to mitigate these limitations by utilizing multiple different PI3K inhibitors (*Figure 1B–D*), ensuring a dose–response (*Figure 1—figure supplement 1A–I, N, O*), directly testing the role of known off-targets (*Figure 1—figure supplement 3*) and utilizing a dominant-negative construct (*Figure 1E*, *Figure 1—figure supplement 1J–L, P*). However, these complementary experiments have their own limitations and thus we are unable to completely rule-out the role of a combinatorial inhibitory effect on cardiac fusion. We look forward to the creation of a genetic model that will help verify our findings.

Overall, our studies delineate a role for the PDGF–PI3K pathway in the mechanisms by which myocardial precursors sense and respond to extracellular signals to move into a position to form the heart. These mechanisms are likely relevant to other organ progenitors including endothelial precursors, endodermal progenitors, and neuromasts; all of which must move from their location of specification to a different location for organ formation. Although varying in their morphogenesis, many of these movements are collective in nature. Indeed, a similar Pdgfra–PI3K signaling cassette is important in the collective directional migration of several organ progenitors including the migration of mesoderm and neural crest cells (*Montero et al., 2003*; *Yang et al., 2008*; *McCarthy et al., 2013*; *Bahm et al., 2017*; *Symes and Mercola, 1996*; *Klinghoffer et al., 2002*; *He and Soriano, 2013*; *Nagel and Winklbauer, 2018*). RTK–PI3K pathways are also important across several cardiac developmental processes, including epicardial development, cardiac neural crest addition, cardiomyocyte growth, cardiac fibroblast movement, and cardiomyocyte contraction (*Crackower et al., 2002*; *Kim et al., 2010*; *Sato et al., 2011*; *Ivey et al., 2019*; *McMullen et al., 2003*; *Shioi et al., 2000*). Similarly, PDGF–PI3K and more generally RTK–PI3K signaling cassettes are activated in several diseases including glioblastomas, gastrointestinal stromal tumors, and cardiac fibrosis (*Wang et al., 1997*; *Cheng et al., 2009*; *Fan et al., 2014*; *Lennartsson et al., 2005*). Thus, the role of this RTK–PI3K cassette in sensing and responding to extracellular signals is likely to be broadly relevant to the etiology of a wide array of developmental processes as well as congenital diseases.

## Materials and methods

**Key resources table**

| Reagent type (species) or resource | Designation | Source or reference | Identifiers | Additional information |
|---|---|---|---|---|
| Genetic reagent (*Danio rerio*) | *Tg(myl7: eGFP)*<sup>twu34</sup> | *Huang et al., 2003* | twu34; RRID: ZFIN_ZDB-GENO-050809-10 | Transgenic |
| Genetic reagent (*Danio rerio*) | *Tg(sox17:eGFP)*<sup>ha01</sup> | *Mizoguchi et al., 2008* | ha01; RRID: ZFIN_ZDB-GENO-080714-2 | Transgenic |
| Genetic reagent (*Danio rerio*) | *Tg(hsp70l:pdgfaa-2A-mCherry;cryaa:CFP)*<sup>sd44</sup> | *Bloomekatz et al., 2017* | sd44; ZDB-ALT-170425-5 | Transgenic |
| Genetic reagent (*Danio rerio*) | *ref (pdgfra*<sup>sk16</sup>) | *Bloomekatz et al., 2017* | sk16; ZDB-ALT-170329-1 | Mutant |
| Genetic reagent (*Danio rerio*) | *ptena*<sup>hu1864</sup> | *Faucherre et al., 2008* | hu1864; ZDB-ALT-080910-1 | Mutant |
| Genetic reagent (*Danio rerio*) | *ptenb*<sup>hu1435</sup> | *Faucherre et al., 2008* | hu1435; ZDB-ALT-080910-2 | Mutant |
| Genetic reagent (*Danio rerio*) | *Tg(myl7:dnPI3K; Cryaa:CFP)* | This paper | | Transgenic, see Materials and methods |
| Genetic reagent (*Danio rerio*) | *Tg(myl7:lck-emgfp)* | This paper | | Transgenic, see Materials and methods |
| Recombinant DNA reagent | pBSRN3-Δp85 | *Carballada et al., 2001* | | |
| Other | myl7 | *Yelon et al., 1999* | ZDB-GENE-991019-3 | mRNA probe |
| Other | axial | *Strähle et al., 1993* | ZDB-GENE-980526-404 | mRNA probe |
| Other | hand2 | *Yelon et al., 2000* | ZDB-GENE-000511-1 | mRNA probe |
| Chemical compound, drug | LY294002 | Millipore-Sigma | Cat# 154447-36-6 | |
| Chemical compound, drug | Dactolisib | Millipore-Sigma | Cat# 915019-65-7 | |
| Chemical compound, drug | Pictilisib | Millipore-Sigma | Cat# 957054-30-7 | |
| Chemical compound, drug | Rapamycin | Selleckchem | Cat# S1039 | |
| Chemical compound, drug | VO-Ohpic trihydrate | Selleckchem | Cat# S8174 | |
| Antibody | phospho-AKT (Rabbit monoclonal) | Cell Signaling | Cat# 4060, RRID: AB_2315049 | WB(1:2000) |
| Antibody | pan-AKT (Rabbit monoclonal) | Cell Signaling | Cat# 4691, RRID: AB_915783 | WB (1:2000) |
| Antibody | Anti-rabbit IgG HRP-conjugated (Goat polyclonal) | Cell Signaling | Cat# 7074, RRID: AB_2099233 | WB (1:5000) |
| Antibody | anti-GFP (Chicken polyclonal) | Abcam | Cat# ab13970, RRID: AB_300798 | IF (1:1000) |
| Antibody | anti-ZO-1 (Mouse monoclonal) | Thermo Fisher Scientific | Cat# 33–9100, RRID: AB_87181 | IF (1:200) |
| Antibody | anti-chicken-488 (Goat polyclonal) | Thermo Fisher Scientific | Cat# A32931TR, RRID: AB_2866499 | IF (1:300) |
| Antibody | anti-mouse-647 (Goat polyclonal) | Thermo Fisher Scientific | Cat# A32728, RRID: AB_2633277 | IF (1:300) |
| Commercial assay or kit | Cell Death detection kit, TMR red | Millipore Sigma | Cat# 12156792910 | |
| Commercial assay or kit | Click-&-Go Cell Proliferation Assay Kit | Click Chemistry Tools | Cat# 1328 | |
| Software, algorithm | mTrackJ | *Meijering et al., 2012* | | ImageJ Plugin for motion tracking and analysis |
| Software, algorithm | Correct 3D Drift | *Parslow et al., 2014* | | ImageJ Plugin for sample drift correction |

| Reagent type (species) or resource | Designation | Source or reference | Identifiers | Additional information |
|---|---|---|---|---|
| Software, algorithm | Prism 10.0.2 | GraphPad | RRID: SCR_002798 | Data visualization and statistics software |
| Software, algorithm | Leica | LASX | Leica Application Suite X, RRID: SCR_013673 | Microscope image processing software |

## Material availability

Materials not available commercially are available upon request to Dr. Joshua Bloomekatz.

## Zebrafish husbandry, microinjections, and plasmid construction

All zebrafish work followed protocols approved by the University of Mississippi IACUC (protocol #21-007). Wildtype embryos were obtained from a mixed zebrafish (*Danio rerio*) AB/TL background. The following transgenic and mutant lines of zebrafish were used: *Tg(myl7: eGFP)^twu34* (RRID: ZFIN_ZDB-GENO-050809-10) (*Huang et al., 2003*), *Tg(sox17:eGFP)^ha01* (ZFIN_ZDB-GENO-080714-2) (*Mizoguchi et al., 2008*), *Tg(hsp70l:pdgfaa-2A-mCherry;cryaa:CFP)^sd44* abbreviated *hs:pdgfaa* (ZDB-ALT-170425-5), *ref (pdgfra^sk16)* (ZDB-ALT-170329-1) (*Bloomekatz et al., 2017*), *ptena^hu1864* (ZDB-ALT-080910-1), and *ptenb^hu1435* (ZDB-ALT-080910-2) (*Faucherre et al., 2008*). All embryos were incubated at 28.5°C unless otherwise noted. Transgenic *Tg(myl7:dnPI3K; Cryaa:CFP)* or *Tg(myl7:lck-emgfp)* F0 founders were established using standard Tol2-mediated transgenesis (*Fisher et al., 2006*). F0 founder pairs were screened by intercrosses looking for a high percentage of F1 embryos with CFP+ eyes and cardiac edema or emGFP+ hearts, respectively. For *Tg(myl7:dnPI3K; Cryaa:CFP)*, stable transgenic lines could not be propagated due to loss of viability. Based on the germline mosaicism of the F0 parents, only a proportion of the F1 embryos are expected to have the transgene. Embryos from four different F0 pairs were analyzed for cardiac fusion phenotypes. Due to germ-line mosaicism, F1 embryos were genotyped after in situ hybridization for the presence of the transgene using standard PCR genotyping. Primer sequences are provided in *Table 1*. *ref (pdgfra^sk16)*, *ptena^hu1864*, and *ptenb^hu1435* were genotyped as outlined in *Bloomekatz et al., 2017*; *Jung et al., 2021*, respectively.

Truncated p85 (dnPI3K) capped mRNA was synthesized from the pBSRN3-Δp85 construct (*Carballada et al., 2001*) and injected at the one-cell stage. To mosaically label cells in the myocardium for protrusion imaging or PI3K-reporter activity, *myl7:lck-eGFP* (30 ng/µl) or *myl7:PH-mkate2* (50 ng/µl) plasmid DNA was injected along with Tol2 transposase (40 ng/µl) into *Tg(myl7:eGFP)* or *Tg(myl7:lck-emgfp)* heterozygous embryos, respectively, at the one-cell stage and embryos were subsequently allowed to develop at 28.5°C.

Plasmids were constructed by using Hifi assembly (NEB, E2621) to transfer lck-eGFP (*Chertkova et al., 2017*) or a truncated version of p85 (*Carballada et al., 2001*) into the middle-entry vector of the tol2 gateway system (*Kwan et al., 2007*), which were verified by sequencing. Primer sequences are provided in *Table 1*. Then gateway recombination between p5E-myl7 promoter, the constructed middle-entry clones, p3E-polyA and either pDESTTol2pA2 (*Kwan et al., 2007*) or

**Table 1.** Primers for genotyping and cloning.

| | Name | Sequence (5′–3′) |
|---|---|---|
| Primers to screen for the *Tg(myl7:dnPI3K)* transgene in F1 embryos | dnPI3K_F1 | GCGGGAAGAGGACATTGACT |
| | dnPI3K_R1 | GCGGGAAGAGGACATTGACT |
| Primers to clone lck-emGFP into the middle-entry vector of the tol2 gateway system | Hi_lck_1F | CAGTCGACTGGATCCGGTACAGATCCGCTAGCCACCATG |
| | Hi_lck_1R | CAGTCGACTGGATCCGGTACAGATCCGCTAGCCACCATG |
| | Hi_emgfp_2F | GGTCGCCACCGTGTCCAAGGGCGAGGAG |
| | Hi_emgfp_2R | GGTCGCCACCGTGTCCAAGGGCGAGGAG |
| Primers to replace actc1b promoter with myl7 promoter in Addgene plasmid 109501 | Hi_CbPHmkate2_F | GGCTGAAAAGCAATCCTGCAGTGACCAAAGCTTAAATCAGTTG |
| | Hi_CbPHmkate2_R | CTCTCCAGAATCACTGCGGCCATGGCCATGGTGGCTACGGATC |

pDESTTol2pA4-Cryaa:CFP (*Bloomekatz et al., 2017*) was used to produce the plasmids *myl7:lck-eGFP* or *myl7:dnPI3K; Cryaa:CFP*, respectively. The *myl7:PH-mkate2* plasmid was created by using Hifi assembly to replace the *actc1* promoter with the *myl7* promoter in the addgene plasmid #109501 (*Hall et al., 2020*).

### Inhibitor treatments

The following inhibitors were used: LY294002 (LY, Millipore-Sigma 154447-36-6), Dactolisib (Dac, Millipore-Sigma 915019-65-7), Pictilisib (Pic, Millipore-Sigma 957054-30-7), Rapamycin (Rap, Selleckchem S1039), and VO-Ohpic trihydrate (VOOH, Selleckchem S8174). For each treatment, inhibitors were freshly diluted serially from stocks such that the same percentage (0.1%) of DMSO was added to 1× E3 in glass vials. 0.1% DMSO was used as a control. 15 dechorionated embryos per vial were incubated in the dark at 28.5°C. In the course of these studies, we noticed that incubation with pharmacological PI3K inhibitors caused a delay in trunk elongation and somite formation along with defects in cardiac fusion (*Figure 1—figure supplement 2*). To ensure our analysis was not obfuscated by a developmental delay, we used somite number to stage match embryos. PI3K-inhibited embryos thus develop approximately 2–3 hr longer than DMSO-treated embryos, prior to analysis.

### Immunoblot, immunofluorescence, in situ hybridization

Embryos at 22s were prepared for immunoblots by deyolking (*Purushothaman et al., 2019*). Primary and secondary antibodies include anti-phospho-AKT (1:2000, Cell Signaling 4060, RRID: AB_2315049), anti-pan-AKT (1:2000, Cell Signaling 4691, RRID: AB_915783), and anti-rabbit HRP conjugated (1:5000, Cell Signaling 7074, RRID: AB_2099233). To identify *pdgfra/ref* heterozygous and homozygous embryos, embryo trunks were clipped and genotyped as described (*Bloomekatz et al., 2017*). The body of the embryo including the heart was snapped frozen and stored at −80°C. After genotyping, embryos were pooled via their genotype and analyzed via immunoblot. To activate Pdgfra, embryos expressing the *Tg(hsp70l: pdgfaa-2A-mCherry)* transgene were heat-shocked at bud stage as described (*Bloomekatz et al., 2017*) and collected at 22s. pAKT and AKT immunoblots were visualized (Azure 600 Imaging system, Azure Biosystems) and quantified using ImageJ (*Stael et al., 2022*). pAkt to Akt ratios were normalized to DMSO.

Immunofluorescence analysis was performed on transverse sections using standard cryoprotection, embedding, and sectioning (*Garavito-Aguilar et al., 2010*). Primary, secondary antibodies and dyes include: anti-GFP (1:1000, Abcam ab13970, RRID: AB_300798), anti-ZO-1 (1:200, Thermo Fisher Scientific 33-9100, RRID: AB_87181), donkey anti-chicken-488 (1:300, Thermo Fisher Scientific A32931TR, RRID: AB_2866499), and donkey anti-mouse-647 (1:300, Thermo Fisher Scientific A32728, RRID:AB_2633277). TUNEL was performed using the TMR red Cell Death detection kit (Millipore Sigma 12156792910). Addition of DNaseI was used to confirm we could detect apoptotic cells. EdU staining was adapted from *Kimmel and Meyer, 2010*; *Schindler et al., 2014* using the Click-&-Go Cell Proliferation Assay Kit (Click Chemistry Tools 1328), with the following modifications: dechorionated 16 hpf embryos were incubated in 0.5 mM EdU for 1 hr at 4°C, rinsed in E3 media multiple times, and then incubated at 28.5 C till 22s.

In situ hybridization was performed using standard protocols (*Alexander et al., 1999*), with the following probes: *myl7* (ZDB-GENE-991019-3), *axial* (ZDB-GENE-980526-404), and *hand2* (ZDB-GENE-000511-1). Images were captured with Zeiss Axio Zoom V16 microscope (Zeiss) and processed with ImageJ.

### Fluorescence imaging

To analyze cardiac fusion (*Figure 1A'–E'*), *Tg(myl7:eGFP)* embryos were fixed, manually deyolked and imaged with a Leica SP8 X microscope. To analyze the anterior endoderm (*Figure 2—figure supplement 1E–G*), *Tg(sox17: eGFP)* embryos were fixed and imaged with an Axio Zoom V16 microscope (Zeiss).

For live imaging, *Tg(myl7:eGFP)* embryos were exposed to DMSO or 20 μM LY at bud stage and mounted at 12 somite stage as described (*McCann et al., 2022*). Mounted embryos were covered with 0.1% DMSO/20 μM LY in Tricaine-E3 solution and imaged using a Leica SP8 X microscope with a HC PL APO ×20/0.75 CS2 objective in a chamber heated to 28.5°C. GFP and brightfield stacks were collected approximately every 4 min for 3 hr. After imaging, embryos were removed from the

mold and incubated for 24 hr in E3 media at 28.5°C. Only embryos that appeared healthy 24 hrs post imaging were used for analysis. The tip of the notochord was used as a reference point to correct embryo drift in the Correct 3D direct ImageJ plugin (*Parslow et al., 2014*). Embryos were handled similarly for imaging protrusions and PH-mkate2 localization, except 15 confocal slices of 1 µm thickness were collected every 1.5 min or 3.5 min, respectively, with a HC PL APO ×40/1.10 CS2 objective.

## Image analysis

To analyze developmental delay, embryonic length (*Figure 1—figure supplement 2*) was measured from the anterior tip of the head to the posterior tip of the tail of each embryo using the free-hand tool of ImageJ. To analyze the anterior endoderm, endoderm width was measured at a point that was 300 µm anterior to where the two sides of the endoderm intersect. The distance between the *hand2* expressing domains was measured at three equidistant positions (~200 µm apart) along the anterior–posterior axis. *Tg(myl7:eGFP)+* cardiomyocytes were counted from blinded and non-blinded 3D confocal images of 20s embryos from four biological replicates using the cell counter addon in ImageJ. No difference between the blinded and non-blinded replicates was detected. Cell proliferation index was calculated as *Tg(myl7:eGFP)+*; EdU+ cells divided by the total number of *Tg(myl7:eGFP)+* cells.

For live imaging of cell movements – the mTrackJ addon in ImageJ (*Meijering et al., 2012*) was used. 20–25 cells per embryo whose position could be determined at each timepoint were chosen from the two most medial columns of myocardial cells on each side of the embryos. From these tracks, cell movement properties including overall displacement, velocity (displacement/time), speed (distance/time), efficiency (displacement/distance), and direction (atan(abs($\Delta y$)/$\Delta x$) × 57.295) – medial–lateral and angular, were calculated. Rose plots and graphs in *Figure 3G, H* consider angular movement, along the anterior–posterior axis, irrespective of its direction. In these plots, individual cells are grouped into 6 bins based on their net direction of movement; the length of each radial bar represents the percentage of cells in each bin.

To quantitate the fluorescent intensity of lck-eGFP and PH-mkate2 – the straight-line function in ImageJ was used to draw a line extending perpendicular from the middle of the most intense region of mkate2 enrichment to the end of the cell. The fluorescence intensity along this line for both eGFP and mkate2 was measured using the plot profile addon.

For live imaging of myocardial membrane protrusions – stacks were processed in Leica LAS X and/or Imaris Viewer (Bitplane) to position the medial edge to the right of the image. Videos of the myocardium were inspected frame by frame in ImageJ for a protrusion. Only cells that were not neighbored by other labeled cells on their medial and lateral edges were analyzed. Each protrusion was classified as either thin (longer than wide) or wide (wider than long). The direction of a protrusion was measured using the 'straight line' function to draw a line from the center of the bottom of the protrusion to the tip. All protrusions of each cell over the entire recording were measured. Graphs, cartoons, and figures were created with Prism (GraphPad), Excel (Microsoft), and Indesign (Adobe).

## Statistics and replication

All statistical analysis was performed in R or Prism (GraphPad). In R the following functions were used T.test(), TukeyHSD(), and fisher.test(). Sample sizes were determined based on prior experience with relevant phenotypes and standards within the zebrafish community. Deviation from the mean is represented as standard error mean or box-whisker plots. In box-whisker plots, the lower and upper ends of the box denote the 25th and 75th percentile, respectively, with a horizontal line denoting the median value and the whiskers indicating the data range. All results were obtained from at least three separate biological replicates, blinded and non-blinded. All replicates are biological. Samples were analyzed before biological sex is determined (*Wang et al., 2007*). Raw data and full p-values included in the source files.

## Acknowledgements

We thank members of the Bloomekatz lab and S Liljegren, B Jones, K Willett, M Jekabsons, Y Qiu for helpful discussions; R Cao and G Roman in the GlyCORE imaging core (NIH-P20GM103460), C Thornton, R Knerr, and P Bolton for animal support as well as C Chang, D Dong, K Kwan, R Parton for providing reagents.

## Additional information

### Funding

| Funder | Grant reference number | Author |
|---|---|---|
| American Heart Association | 18CDA34080195 | Joshua Bloomekatz |
| Eunice Kennedy Shriver National Institute of Child Health and Human Development | R15HD108782 | Joshua Bloomekatz |
| National Institute of General Medical Sciences | P20GM103460 | Joshua Bloomekatz |

The funders had no role in study design, data collection, and interpretation, or the decision to submit the work for publication.

### Author contributions

Rabina Shrestha, Tess McCann, Conceptualization, Data curation, Formal analysis, Investigation, Visualization, Methodology, Writing - review and editing; Harini Saravanan, Formal analysis, Investigation, Visualization, Methodology; Jaret Lieberth, Validation, Methodology; Prashanna Koirala, Investigation; Joshua Bloomekatz, Conceptualization, Data curation, Formal analysis, Supervision, Funding acquisition, Investigation, Visualization, Methodology, Writing - original draft, Project administration, Writing - review and editing

### Author ORCIDs

Joshua Bloomekatz ⓘ http://orcid.org/0000-0001-5816-2756

### Ethics

All animals were handled according to protocols approved by the University of Mississippi Institutional Animal Care and Use Committee (IACUC) (protocol #21-007), in accordance with the recommendations of the National Institutes of Health (NIH).

### Decision letter and Author response

Decision letter https://doi.org/10.7554/eLife.85930.sa1
Author response https://doi.org/10.7554/eLife.85930.sa2

## Additional files

### Supplementary files
• MDAR checklist

### Data availability

All data are included in the manuscript and supporting files. Source data files have been provided for all figures.

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
