## [Editor Report]

This is a valuable study that shows the involvement of phosphoinositide 3-kinase (PI3K) signaling downstream of platelet-derived growth factor receptor α in latero-medial migration of cardiomyocytes during the formation of the early heart tube during zebrafish development. The authors provide convincing evidence for the role of PI3K in cardiomyocyte migration using multiple PI3K inhibitory drugs, expression of a dominant negative PI3K subunit, and rescue of the Pdgfaa ligand over-expression phenotype using mild PI3K inhibition, approaches which show strong alignment and which are quantified using live imaging. The demonstration of cardiomyocyte protrusions biased in the direction of migration, and randomised after PI3K inhibition, is a promising area for future exploration.

---

## [Decision Letter]

**Decision letter after peer review:**

Thank you for submitting your article "The myocardium utilizes Pdgfra-PI3K signaling to steer towards the midline during heart tube formation" for consideration by *eLife*. Your article has been reviewed by 3 peer reviewers, one of whom is a member of our board of Reviewing Editors, and the evaluation has been overseen by Didier Stainier as the Senior Editor. The following individual involved in the review of your submission has agreed to reveal their identity: Osvaldo Contreras (Reviewer #2).

Essential revisions:

The reviewers found this to be a valuable contribution as a Research Advance – however, significant weaknesses and discrepancies were identified. The authors therefore should respond to all of the reviewers' comments. Reviewers have asked for additional data to support current findings and to extend the model of PDGFRa/PI3K-mediated chemotactic migration of cardiomyocyte precursors during heart tube formation in zebrafish. We consider essential revisions to be:

1) Address spatially PI3K activity in vivo to further address the model and to confirm and understand the impact of PI3K inhibition using drugs, and the dominant negative reagent.

2) Address the discrepancies between single cardiomyocyte trajectory data in the original and current work, and repeat more rigorously if necessary. Direct the research towards understanding the different behaviors evident in anterior, medial, and posterior zones, and align with the model.

3) Strengthen experimentally the observations around cardiomyocyte protrusions, including experiments that would extend the model by distinguishing between individual cell versus collective cell migration. If possible, address whether protrusions are involved in cell steering.

4) It would be a strong addition to the study if the authors could show rescue of PI3K inhibition (with drugs and/or dominant negative transgene) with the Pdgf-aa transgene used in the previous study and in the current study to support PI3K/pAKT being downstream of PDGFRA.

5) Please include the zebrafish model in the title as per guidelines and please spell out PDGFRa and PI3K.

*Reviewer #1 (Recommendations for the authors):*

The manuscript by Shrestha and Bloomekatz and colleagues addresses the role of PI3K signaling in latero-medial migration of committed cardiomyocyte cell progenitor populations during the formation of the early heart tube in zebrafish development, disruption of which causes complete or partial cardia bifida. The work is submitted as a Research Advance, which builds on a previous Research Article in *eLife* (Bloomekatz et al., *eLife* 2017;6:e21172). In the previous paper, Bloomekatz and colleagues reported the isolation and analyses of the ref mutation, showing that it interrupts the function of the Pdgfra gene. They go on to characterise defects in the medial migration of cardiac progenitors in the genetic mutant and also in pdgfra morpholino knockdown experiments. Pdgfra is expressed in the anterior lateral mesoderm during migration and fusion, whereas its ligand Pdgfaa is expressed in the anterior endoderm, known to be important for cardiomyocyte fusion, potentially for physically propelling (via extracellular matrix) the cardiac mesoderm along. Live imaging and quantification showed that the Pdgfra expression in cardiac mesoderm is required for directional movement (steering) of cardiomyocytes to the midline during heart tube formation, presumably via a chemotactic mechanism involving graded signals from endoderm. Effects on velocity and efficiency of movement of cardiomyocytes were not evident.

In the present manuscript, Bloomekatz and colleagues explore the role of PI3K in latero-medial cardiomyocyte migration in zebrafish, potentially working downstream of Pdgfra. A similar overall approach is taken as in the first paper, albeit that the PI3K is inhibited using the drug LY294002 (confirmed using other classes of inhibitor), or cardiomyocyte-specific transgenic expression of a truncated dominant negative form of PI3K. In terms of prior knowledge, and therefore the novelty of this paper, the PI3K pathway is known to function downstream of Pdgfr signaling and indeed in migration and proliferation of Pdgf-dependent cells and other cell types in the development, including in collective cell migrations. PI3K is also involved in the formation of cell protrusions in some settings, and cardiomyocyte protrusions have been observed in developing mouse hearts. The manuscript is transparent on background studies; however, involvement of PI3K is therefore somewhat expected. Nonetheless, the authors show convincingly that PI3K inhibition affects cardiomyocyte latero-medial migration in a dose-dependent manner. They define a broad window for the involvement of PI3K during the migration process and show that cardiomyocyte-specific expression of dominant negative PI3K causes migration defects in 100% of embryos, demonstrating that this is a cell-autonomous process. Similar to the original study, live imaging with a myl7-eGFP transgene enabled the authors to quantify defects in individual cardiomyocyte trajectories and determine average scores for velocity, efficiency, and direction of movement, which were all compromised after PI3K inhibition. Elements of cell behaviour at different time points along the migration process were considered. As a potential mechanism, they demonstrate cellular protrusions in migrating cardiomyocytes and quantify their number, duration, and direction relative to the axis of migration. An important finding is that these protrusions are oriented towards the direction of migration in control embryos, but this is lost after PI3K inhibition. Finally, an interaction between Pdgfra heterozygosity and PI3K inhibition is demonstrated, suggesting that PI3K does function downstream of Pdgfra signaling.

Recommendations:

1. Inhibition of PI3K might lead to phenotypes that are weaker (partial inhibition) or stronger (inhibition of PI3K in tissues beyond cardiomyocytes); however, one would expect given the results of the cardiomyocyte-specific dnPI3K transgene, that the phenotypes of Pdgfra ref mutants and PI3K inhibition would closely align. One significant concern, therefore, is that the quantifications of cell trajectories look very different comparing Bloomekatz et al. 2017 to the current study. In the original study, control cardiomyocytes have much broader angles of trajectory compared to the very limited angles in the current study. Similarly, ref mutant cardiomyocytes often migrate in alternate directions along the latero-medial axis and sometimes in a net backward direction, not seen in the current study with PI3K inhibition. Comparing the trajectories in the current paper, it appears that they are mostly latero-medial in orientation anteriorly, whereas those occurring posteriorly have a distinct anterior-medial direction. These could be driven by different processes. Visual inspection of the trajectories in the inhibited embryos seems to show that the posterior type is preserved, whereas the anterior type is diminished. The authors have perhaps lost the opportunity to segment classes of behavior to refine the model of cardiomyocyte chemotactic morphogenesis. The authors do present segmentation of anterior versus posterior behaviors; however, it seems that the posterior ones differ between the control and mutant, not the anterior ones, which might go against the visual inspection of panels in both Figure 3B, D, and Suppl. Figure 5D. At a minimum, the authors should address the differences between the two studies – are these differences a product of methodology or normalisation (e.g. of notochord contraction) or to features of the gene model, if so which? and how does this relate to the model of migration? It would help to better relate trajectory findings to literature models.

2. It would strengthen the paper to show spatially PI3K activity across the migration period. Phospho-ATK has been used previously to show receptor tyrosine kinase activity in vivo in different settings (as shown in this study by western blotting).

3. The characterization of cardiomyocyte epithelial morphology is cursory, and I think other markers of epithelial morphology, apical-basal polarity or integrity could be shown, perhaps N-CAM. The authors should be more specific in their description of the processes leading to the second dorsal layer (which is currently vague and seems not visible in the control sample so panels are not comparable).

4. Are the protrusions at the leading edge of the cardiomyocyte population or throughout? Does this instruct on how migration occurs – as individual cells in a partial mesenchymal state, or as a sheet?

5. The tools used to explore endodermal integrity (gross expression of axial and Sox17-gfp) seem inadequate. The question as to whether endoderm is compromised is paramount, as endoderm has been proposed to instruct cardiomyocyte migration, likely by the provision of chemotactic ligands (e.g. Pdgf-aa), however, potentially also by physical means. Is its own migration relative to that of cardiomyocytes compromised? Knowing this could advance the model.

6. Page 6. Examples of control versus inhibited embryos stage-matched for somite number should be shown in Suppl. Figure 1. I don't see any reference to somite stage matching in Suppl. Figure 2, only that somite number is retarded in inhibited embryos.

7. Please state in the manuscript and/or show in a table the known specificities of the inhibitors used.

8. I think the statistics for the cell behavioural studies should be shown on the graphs in a more conventional way.

9. It would be helpful to cite other examples where cardiomyocyte protrusions have been described; for example, dedifferentiated neonatal cardiomyocytes in vitro (Eldad Tzahor lab), regenerating scar after myocardial infarction (James Martin lab) and myocardialization of outflow track cushions (Antoon Moorman lab).

*Reviewer #2 (Recommendations for the authors):*

Recommendations:

1. The authors should refer to their concentrations used in the results field (Page 6). I also think that the concentrations used are higher than expected (especially for Dactolisib and Pictilisib). Thus, the authors should be extremely cautious about their results and their interpretations since off-target effects could potentially target myocardial fusion. I had a quick look and the IC50 of Pictilisib is in nM range and the authors used 50µM, which is 1000-fold higher.

2. I suggest the authors extend their evaluation of PI3K downstream targets including GSK3B and mTOR. This aims to better understand the mechanisms associated with PI3K pharmacological inhibition.

3. The authors mention "Furthermore, the PI3K signaling pathway is known to promote cell proliferation and cell survival (29) however, we did not find a difference in the number of cardiomyocytes in DMSO- or LY- treated embryos at 20s (Suppl. Figure 3G-I). I suggest the authors evaluate not only the total cardiomyocyte number but cardiomyocyte cell cycle and/or proliferation, using techniques such as BrdU/EdU uptake, Ki67 labelling, and phosphorylated Histone H3 in Serine 10. Equal cardiomyocyte numbers do not necessarily point to no defects in the cardiomyocyte cell cycle and proliferation potentially induced by PI3K inhibitors (known to affect the cell cycle in many different cell types and contexts).

4. The authors refer to cardiomyocyte apoptosis on Page 7 but no figure reference exists. Please, check and add the missing results.

5. Line 19, page 8. The authors refer to myocardial movement, but they have not directly addressed that point yet. I suggest the authors amend the sentence. The same applies to line 10, page 10.

6. Line 1, page 10. I would change "indicating" to "suggesting".

7. Would the increased PDGFRa activity obtained using mutant zebrafish over-expressing Pdgf-aa rescue PI3K-inhibited embryos? I think this is an important point to be addressed.

8. Page 16, line 14. The authors refer to "protrusion formation" but if I understand well, they never evaluated that later cellular effect.

Recommendations related to Figures and related conclusions:

1. Immunofluorescence against phosphorylated AKT during cardiac fusion (using Tg(myl7:egfp zebrafish)) could strengthen the authors' results.

2. The authors should evaluate PI3K activity (AKT phosphorylation) in the case of Dactolisib and Pictilisib inhibitors. dnPI3K would also be highly appreciated.

3. In order to explain the PI3K-mediated defects during cardiac fusion, I would change the order of Figure 4 by Figure 3. Thus, PI3K-induced defects on myocardial membrane protrusions may better explain the resultant medial movement and myocardium speed during cardiac fusion. It reads better from my point of view in that way.

4. The authors should quantify the number of protrusions per cell and not only the direction. If the number of protrusions is reduced that should open new avenues related to the coordinated migratory mechanisms during cardiac fusion, but if the numbers do not change that may reinforce the authors conclusions.

5. Figure 5F. The results of 10µM LY inhibitor in Pdgfra +/+ embryos do not match well with Supplementary Figure 1M. Please check and discuss if necessary.

6. Do the authors observe a genetic interaction between dnPI3K and Pdgfras16 mutants (het or homo)?

7. Figure 5. Western blots for p-AKT are based on total embryo lysates. Thus, it is expected to find what the authors found related to Pdgfra mutants and hs:pdgfaa. The manuscript would be strengthened if the authors could provide direct evidence of phosphorylated AKT in the myocardial region during cardiac fusion. The authors could use immunofluorescence and confocal imaging.

8. I am a bit conflicted with Supplementary Figure 5C. What each dot represents? A single cell? I suggest the authors read the latest thread about some important statistical considerations:

https://doi.org/10.1083/jcb.202001064

https://rupress.org/jcb/article/219/6/e202001064/151717/SuperPlots-Communicating-reproducibility-and

*Reviewer #3 (Recommendations for the authors):*

Overall, the manuscript is well written and experiments are well controlled providing sufficient evidence to substantiate most of their conclusions. There are a number of questions that need to be addressed by the authors.

Specific points:

1. What is interesting is that the CPCs have some intrinsic migration that is independent of the endoderm. The finding that Pi3K is required for the cardiac fusion process is not too surprising as the authors showed in their previous work that PDGF signaling is driving this process and PI3K is a well-known transducer of PDGF signaling. An important question remaining is what type of cell migration the CPCs undergo to fuse at the midline. Are the CPCs migrating as individual cells that all respond to the PDGF from the midline or do the CPCs migrate collectively and only a few respond to the PDGF signal? The authors seem to prefer the collective migration of CPCs but it seems that this has not been tested experimentally. Testing this directly would significantly increase the impact of this study. With the myl7:dnPI3K construct the authors seem to have a good tool to test this now. This could be addressed by analysing embryos in which the cardiac field contains a mixture of wt CPCs and CPCs expressing the dnPIK3 for example by cell transplantation experiments in which cells from embryos derived from the myl7:dnPI3K F1 fish are transplanted to wt embryos or by DNA injections resulting in mosaic expression. It would be very interesting to see how CPCs expressing the dnPI3K behave when surrounded by wt cells. If collective migration is indeed driving cardiac fusion one would expect that these dnPI3K cells behave like wt cells.

2. In Figure 3 the authors show that migration of CPCs treated with LY is affected. They plotted the CPC velocities in 3E and here it looks as if fast-migrating CPCs are less affected compared to slower-migrating CPCs. Fast migration is mostly seen in the posterior part of the bilateral heart fields, so the question arises whether CPCs along the anterior-posterior axis is affected differently by the LY treatment.

3. In fig3E the authors show the total velocity of the CPCs in all directions. As they conclude at the end that PDGF-PI3K signaling mostly affects the medial cell migration, it would be important to analyse the velocities of the CPCs only in the medial-lateral direction.

4. In Figure 3G the authors show the angles at which the CPCs migrate based on the time lapses. In Figure 3B the tracks show that some cells (mostly located anteriorly) move in a medial-posterior direction while other cells (mostly located in the posterior part) move in an anterior-medial direction. The rose plot however displays only positive angles (medial-anterior direction) and no angles in the medial-posterior direction. Please explain why this was done. In addition, the angles shown here for wt embryos (average 30 degrees) seem very different from the angles reported for wt embryos in their previous work (Fig6L, average around 60 degrees), which look more like the angles reported here for the LY treated embryos (average around 60 degrees). Please explain this.

5. In Figure 4 the authors analyse the direction and persistence of cellular protrusion in CPCs. Especially the direction of these protrusions seems affected by LY suggesting that PDGF from the midline may be responsible for this phenomenon. It would therefore be very interesting to test this in their hs:pdgf line by analysing protrusion direction when pdgf is produced by all cells. If PDGF from the midline is responsible for their direction, overexpressing pdgf in all cells should give a similar effect as inhibiting PI3K.

---

## [Author Response]

Essential revisions:The reviewers found this to be a valuable contribution as a Research Advance – however, significant weaknesses and discrepancies were identified. The authors therefore should respond to all of the reviewers' comments. Reviewers have asked for additional data to support current findings and to extend the model of PDGFRa/PI3K-mediated chemotactic migration of cardiomyocyte precursors during heart tube formation in zebrafish. We consider essential revisions to be:1) Address spatially PI3K activity in vivo to further address the model and to confirm and understand the impact of PI3K inhibition using drugs, and the dominant negative reagent.

We thank the reviewers for this suggestion. We agree that this is an important issue. We’ve addressed the issue of tissue-specificity of PI3K activity in a couple different ways. For example, by expressing a dominant negative construct specifically in the myocardium using the *myl7* promoter (Figure 2C-H), we showed that PI3K signaling is specifically required within myocardial cells to regulate cardiac fusion.

We have further addressed this question in our revisions by employing a pleckstrin homology (PH)-domain PIP3 reporter, *Tg(myl7:PH-mkate2)*. PH domains from several proteins including from BTK have been shown to translocate to the membrane in response to PI3K-induced PIP3 production (Hall, TE et al. 2020, PMID: 32709891; Yoo, SK et al. 2010, PMID: 20159593). In DMSO-treated embryos, we observed PH-mkate2 localize to the plasma membrane of myocardial cells, indicating PI3K activity. Conversely, we found that the PH-mkate2 in the cytoplasm or in sub-cellular compartments in PI3K-inhibited embryos. A detailed analysis of these new results is contained within Figure 2 —figure supplement 2 as well as to changes in the text (pages 11-12, lines 19-22, 1-6, respectively).

2) Address the discrepancies between single cardiomyocyte trajectory data in the original and current work, and repeat more rigorously if necessary. Direct the research towards understanding the different behaviors evident in anterior, medial, and posterior zones, and align with the model.

We thank the reviewers for their close reading of our manuscript and the literature. In our study, myocardial cell trajectories in DMSO-treated embryos are initially dominated by medial-lateral directionality and then by more angular movement (anterior-posterior directionality) (Figure 3A-B). These observations are overall consistent with findings from both the original work (Bloomekatz et al. 2017) and from other previously published studies (Holtzman et al. 2007). As suggested by the reviewers, we have increased our analysis of myocardial movements in this manuscript separating them by developmental stage (early, posterior, anterior) and by anterior, medial and posterior zones (Figure 3 —figure supplement 2C-F). These analyses reaffirm our qualitative observations of increasing angular movements during later developmental stages and of relative increases in angular velocity in anterior and posterior cells.

In this study a greater percentage of tracks are medially directed (Figure 3G) when compared to Bloomekatz et al. 2017 – Figure 6L. These minor differences are likely a result of technical differences in experimentation, including normalization using the notochord and differences in the developmental stages captured by the timelapse data. Variation between could also contribute these differences. We have incorporated these considerations into our revised discussion (page 18, lines 16-20).

3) Strengthen experimentally the observations around cardiomyocyte protrusions, including experiments that would extend the model by distinguishing between individual cell versus collective cell migration. If possible, address whether protrusions are involved in cell steering.

We share the reviewers’ enthusiasm for our observations of cardiomyocyte protrusions. In our studies we identified myocardial protrusions during cardiac fusion which are oriented in the medial-lateral direction in DMSO-treated embryos but are mis-oriented in PI3K-inhibited embryos. In our revisions we have strengthened our observations of these myocardial protrusions, as suggested by the reviewer. Through these enhanced observations we identified different protrusion types: thin protrusions similar to filopodia and wide protrusions similar to pseudopodia. And we analyzed how the ratio and orientation of these different protrusions changes in LY-treated (Figure 4—figure supplement 1).

We do agree with the reviewers regarding the value of further experiments regarding the overall role for protrusions in cardiomyocyte movement and their role in the collective movement of the myocardium. However, these experiments are distinct from our current study focusing on PI3K signaling and its connection to Pdgfra. We thus plan to include them in a more comprehensive study of the role of cardiomyocyte protrusions in cardiac fusion in the future.

4) It would be a strong addition to the study if the authors could show rescue of PI3K inhibition (with drugs and/or dominant negative transgene) with the Pdgf-aa transgene used in the previous study and in the current study to support PI3K/pAKT being downstream of PDGFRA.

In response to valuable reviewer feedback regarding strengthening our finding of an interaction between Pdgfra and PI3K signaling during cardiac fusion, we investigated if the cardiac fusion defects found in embryos overexpressing *pdgfaa* could be rescued by using a sub-phenotypic dose of LY. We found that while 55.5% of heat-shocked *Tg(hs:pdgfaa)* embryos exposed to DMSO display cardiac fusion defects, only 17% of heat-shocked *Tg(hs:pdgfaa)* embryos exposed to 10µM LY showed cardiac fusion defects (p-value 0.0184) (Figure 5G-J). This result complements our previous findings of genetic interaction between *pdgfra -/+* and 10µM LY exposure (Figure 5C-F) and that PI3K activity is regulated by changes in *pdgfra* activity (both loss- and gain- offunction) (Figure 5A, B). Together these experiments suggest that Pdgfra and PI3K signaling work together to regulate cardiac fusion. These changes have been incorporated into Figure 5 and page 16, lines 3-5.

5) Please include the zebrafish model in the title as per guidelines and please spell out PDGFRa and PI3K.

Thank you for pointing out this oversight, it has been corrected.

Reviewer #1 (Recommendations for the authors):Recommendations:1. Inhibition of PI3K might lead to phenotypes that are weaker (partial inhibition) or stronger (inhibition of PI3K in tissues beyond cardiomyocytes); however, one would expect given the results of the cardiomyocyte-specific dnPI3K transgene, that the phenotypes of Pdgfra ref mutants and PI3K inhibition would closely align. One significant concern, therefore, is that the quantifications of cell trajectories look very different comparing Bloomekatz et al. 2017 to the current study. In the original study, control cardiomyocytes have much broader angles of trajectory compared to the very limited angles in the current study. Similarly, ref mutant cardiomyocytes often migrate in alternate directions along the latero-medial axis and sometimes in a net backward direction, not seen in the current study with PI3K inhibition. Comparing the trajectories in the current paper, it appears that they are mostly latero-medial in orientation anteriorly, whereas those occurring posteriorly have a distinct anterior-medial direction. These could be driven by different processes. Visual inspection of the trajectories in the inhibited embryos seems to show that the posterior type is preserved, whereas the anterior type is diminished. The authors have perhaps lost the opportunity to segment classes of behavior to refine the model of cardiomyocyte chemotactic morphogenesis. The authors do present segmentation of anterior versus posterior behaviors; however, it seems that the posterior ones differ between the control and mutant, not the anterior ones, which might go against the visual inspection of panels in both Figure 3B, D, and Suppl. Figure 5D. At a minimum, the authors should address the differences between the two studies – are these differences a product of methodology or normalisation (e.g. of notochord contraction) or to features of the gene model, if so which? and how does this relate to the model of migration? It would help to better relate trajectory findings to literature models.

We thank the reviewer for their detailed reading of our manuscript and our previous work. As we outlined in our response to essential revisions #2, we have increased our analysis of cardiomyocyte trajectories in wild-type/DMSO-treated embryos (Figure 3 —figure supplement 2C-F). Overall, our data is consistent both with our previous work and with the work of others (Holtzman et al. 2007, PMID: 17537802). The main difference between these studies resides in the rose plots of wild-type myocardial trajectories in Bloomekatz et al. 2017 and the DMSO-trajectories in this manuscript. This difference is likely due to experimental variation in the developmental stages over which the timelapse data were collected, differences in normalization to the notochord, and/or differences reflective of variance between embryos. We now include a discussion of these differences in manuscript (page 18, lines 16-20).

In regards to comparisons of cell trajectory between ref/pdgfra mutant and PI3K mutants; there are strong similarities between these trajectories including defects in medial-lateral directionality, speed and movement defects being visible at the start of the timelapse movies. However, as the reviewer notes there are also differences between these trajectories, with the *ref/pdgfra* mutant myocardial cells in a few embryos displaying lateral movement that is not present in PI3K-inhibited embryos. These differences could be due to extent of *pdgfra* inhibition compared to PI3K inhibition. Furthermore, other intracellular signaling pathways parallel to PI3K signaling and downstream of Pdgfra, such as MAPK and PLCg could play minor compensatory roles in steering myocardial movement, thus explaining the less severe result in PI3Kinhibited embryos. We have increased our discussion of these considerations in the revised manuscript (page 19, lines 2 2-8).

The reviewer also mentions examining whether locations in cardiac ring (anterior vs posterior) are differently affected. We have now included this analysis in Figure 3 —figure supplement 2.

2. It would strengthen the paper to show spatially PI3K activity across the migration period. Phospho-ATK has been used previously to show receptor tyrosine kinase activity in vivo in different settings (as shown in this study by western blotting).

We agree with the reviewer that the issue of spatial PI3K activity/tissue-specificity is an important one. As we have highlighted in our response to essential revision #1 – we have addressed this issue in a number of ways including myocardial specific inhibition of PI3K activity (Figure 2). In our revisions we have now included experiments involving a PI3K activity PH-domain reporter (Figure 2 —figure supplement 2) in which we confirm PI3K activity in the myocardium during the later stages of cardiac fusion. In the future we plan to extend these studies to all stages of cardiac fusion.

3. The characterization of cardiomyocyte epithelial morphology is cursory, and I think other markers of epithelial morphology, apical-basal polarity or integrity could be shown, perhaps N-CAM. The authors should be more specific in their description of the processes leading to the second dorsal layer (which is currently vague and seems not visible in the control sample so panels are not comparable).

We thank the reviewer for their comment. We have modified Figure 1—figure supplement 4 to indicate the location of the second dorsal layer (arrows), which has been previously noted (Trinh, LA et al. 2004, PMID: 15030760; Ye, D et al. 2015, PMID: 26329600). And we have clarified that this process occurs around the same time myocardial cells move ventral to the endoderm, a process that has been termed subduction (pages 7-8, lines 21-22, 1-2, respectively), although the cellular and molecular processes that underlie this formation have not been fully elucidated.

4. Are the protrusions at the leading edge of the cardiomyocyte population or throughout? Does this instruct on how migration occurs – as individual cells in a partial mesenchymal state, or as a sheet?

We agree with the reviewer, an examination of how protrusions change in different regions of the myocardial populations and at different developmental timepoints is an interesting question. In our revisions we have enhanced our analysis of cardiomyocyte protrusions by identifying protrusions with different morphologies. In our future work, we hope to build on these studies including addressing the role of these protrusions in migration.

5. The tools used to explore endodermal integrity (gross expression of axial and Sox17-gfp) seem inadequate. The question as to whether endoderm is compromised is paramount, as endoderm has been proposed to instruct cardiomyocyte migration, likely by the provision of chemotactic ligands (e.g. Pdgf-aa), however, potentially also by physical means. Is its own migration relative to that of cardiomyocytes compromised? Knowing this could advance the model.

We appreciate Reviewer 1’s comment regarding investigating endodermal integrity during cardiomyocyte migration. In response to this feedback, we have now added high resolution confocal images of the endoderm in DMSO-treated and PI3K-inhibited embryos to support our conclusion that the morphology of the endoderm is not compromised in PI3K-inhibited embryos (Figure 2 —figure supplement 1I, J).

6. Page 6. Examples of control versus inhibited embryos stage-matched for somite number should be shown in Suppl. Figure 1. I don't see any reference to somite stage matching in Suppl. Figure 2, only that somite number is retarded in inhibited embryos.

All the embryos shown in the figures, including in Suppl. Figure 1 (now – Figure 1 —figure supplement 1) are stage matched by somite number, except the embryos in Suppl. Figure 2 (now – Figure 1 —figure supplement 2) which are matched by time postfertizilation (hpf). We have modified the text (page 6, lines 13-17) to clarify this fact.

7. Please state in the manuscript and/or show in a table the known specificities of the inhibitors used.

We thank the reviewer for their suggestion and we have now included statements and references to known off-targets of each of the inhibitor (page 7, lines 6-8). They are also included in Author response table 1.

**Author response table 1. sa2table1:** 

Inhibitors	Target	Known off-targets
LY294002	Class I PI3Ks	mTOR, DNA-PK
Dactolisib	Class I PI3Ks, mTOR	
Pictilisib	Class I PI3Ks	mTOR
Rapamycin	mTOR	
VOOH	Pten	

References: Maira, SM. et al. 2008 PMID: 18606717; Folkes, A. et al. 2008 PMID:

18754654; Rosivatz, E. et al. 2006 PMID: 17240976, Gharbi, S.I. et al. 2007 PMID:

17302559.

8. I think the statistics for the cell behavioural studies should be shown on the graphs in a more conventional way.

We had previously represented variation in our graphs by showing all individual cells, and displaying a box-whisker plot, which shows the median, the location of the quartiles and the min and max. As suggested by the reviewers we have now switched the graphs in the cell behavioral studies to superplots (Lord, S. et al. 2020 PMID: 32346721) in which all cells are shown along with the averages from individual embryos, and the mean and standard error are shown along with data from the appropriate statistical test, see Figures 3E-H, Figure 3 —figure supplement 2C-F.

9. It would be helpful to cite other examples where cardiomyocyte protrusions have been described; for example, dedifferentiated neonatal cardiomyocytes in vitro (Eldad Tzahor lab), regenerating scar after myocardial infarction (James Martin lab) and myocardialization of outflow track cushions (Antoon Moorman lab).

We thank the reviewer for alerting us to this oversight, these citations as well as those for myocardial protrusions occurring during cardiac trabeculation are now included in the manuscript (page 19, lines 17-19).

Reviewer #2 (Recommendations for the authors):Recommendations:1. The authors should refer to their concentrations used in the results field (Page 6). I also think that the concentrations used are higher than expected (especially for Dactolisib and Pictilisib). Thus, the authors should be extremely cautious about their results and their interpretations since off-target effects could potentially target myocardial fusion. I had a quick look and the IC50 of Pictilisib is in nM range and the authors used 50µM, which is 1000-fold higher.

We share the reviewers concern regarding using high concentrations of inhibitor. However, the reported nanomolar IC50 values are determined using cell-free assays, and may not account for issues of stability, metabolism, penetrance, among others that are encountered in live animal studies. Indeed, our concentrations fall within the range of concentrations used in other live animal studies (Montero, JA et al. 2003, PMID: 12906787; Sasore, T. et al. 2014, PMID: 25144531; Junttila, T. et al. 2009, PMID: 19411071).

2. I suggest the authors extend their evaluation of PI3K downstream targets including GSK3B and mTOR. This aims to better understand the mechanisms associated with PI3K pharmacological inhibition.

We thank Reviewer 2 for this valuable feedback. In response, we investigated if mTOR is required for cardiac fusion by exposing embryos to increasing concentrations of Rapamycin. We did not observe any cardiac fusion defects in embryos when mTOR was inhibited, even at very high concentrations, as shown in Figure 1 —figure supplement 3A-F.

3. The authors mention "Furthermore, the PI3K signaling pathway is known to promote cell proliferation and cell survival (29) however, we did not find a difference in the number of cardiomyocytes in DMSO- or LY- treated embryos at 20s (Suppl. Figure 3G-I). I suggest the authors evaluate not only the total cardiomyocyte number but cardiomyocyte cell cycle and/or proliferation, using techniques such as BrdU/EdU uptake, Ki67 labelling, and phosphorylated Histone H3 in Serine 10. Equal cardiomyocyte numbers do not necessarily point to no defects in the cardiomyocyte cell cycle and proliferation potentially induced by PI3K inhibitors (known to affect the cell cycle in many different cell types and contexts).

We appreciate Reviewer 2’s suggestions and we have now included an experiment investigating the number of cardiomyocytes in S phase by analyzing EdU uptake. We did not find a statistical difference between in DMSO and PI3K inhibited embryos. These findings are reported in Figure 1 —figure supplement 4.

4. The authors refer to cardiomyocyte apoptosis on Page 7 but no figure reference exists. Please, check and add the missing results.

We have now added images (Figure 1 —figure supplement 4M-P) and quantitative data (Figure 1 —figure supplement 4 source data), showing no TUNEL+ cardiomyocytes in DMSO- nor in LY-treated embryos, even though TUNEL+ cells are evident when embryos are treated with DNAase.

5. Line 19, page 8. The authors refer to myocardial movement, but they have not directly addressed that point yet. I suggest the authors amend the sentence. The same applies to line 10, page 10.

We have changed movement to translocation (page 10, line 12; page 12, line 17) as requested by the reviewer.

6. Line 1, page 10. I would change "indicating" to "suggesting".

We have made the suggested change (page 12, line 7), thank you.

7. Would the increased PDGFRa activity obtained using mutant zebrafish over-expressing Pdgf-aa rescue PI3K-inhibited embryos? I think this is an important point to be addressed.

We thank Reviewer 2 for their suggestion. During our revisions, we performed this experiment and found that a sub-phenotypic reduction in PI3K activity could rescue the cardiac fusion defects caused by overexpressing Pdgf-aa. This result is reported in Figure 5G – J and in the manuscript (page 16, lines 3-5).

8. Page 16, line 14. The authors refer to "protrusion formation" but if I understand well, they never evaluated that later cellular effect.

Thank you for pointing out this oversight. We have now modified the text (page 17, line 9).

Recommendations related to Figures and related conclusions:1. Immunofluorescence against phosphorylated AKT during cardiac fusion (using Tg(myl7:egfp zebrafish)) could strengthen the authors' results.

We appreciate the reviewer’s suggestion. However, despite our diligent efforts we have been unable to reliably achieve a successful immunofluorescence staining in zebrafish for phosphorylated Akt (pAkt). As an alternative approach, we employed a pleckstrin homology (PH) domain reporter – PH-mkate2, to assess PI3K activation during cardiac fusion. PH-domains that bind to PIP3 are localized to the membrane when PIP3 is present. Our observations revealed PH-mkate2 at the plasma membrane of cardiomyocytes in DMSO-treated embryos, but not in LY-treated embryos. This information is now included in our revised manuscript (see Figure 2 —figure supplement 2 and pages 11-12, lines 19-22, lines 1-6, respectively).

2. The authors should evaluate PI3K activity (AKT phosphorylation) in the case of Dactolisib and Pictilisib inhibitors. dnPI3K would also be highly appreciated.

We thank the reviewer for the suggestion and apologize for this oversight. We have now evaluated pAKT in comparison to Akt levels in embryos treated with Dactolisib, Pictilisib and dnPI3K. We observed a dose-dependent decrease in PI3K activity as reported by pAKT-to-AKT levels. This data is now included in our revised manuscript (Figure 1 —figure supplement 1Q-S, and on page 7, lines 1-5, 15).

3. In order to explain the PI3K-mediated defects during cardiac fusion, I would change the order of Figure 4 by Figure 3. Thus, PI3K-induced defects on myocardial membrane protrusions may better explain the resultant medial movement and myocardium speed during cardiac fusion. It reads better from my point of view in that way.

We thank the reviewer for this suggestion – having revised the manuscript in this order; the logical flow felt off and so we have changed it back. We hope our other revisions have increased the logical flow for the readers.

4. The authors should quantify the number of protrusions per cell and not only the direction. If the number of protrusions is reduced that should open new avenues related to the coordinated migratory mechanisms during cardiac fusion, but if the numbers do not change that may reinforce the authors conclusions.

We appreciate the reviewer’s suggestion. In response to this query, we quantified the frequency of protrusions per cell per hour and found that cardiomyocytes in DMSOtreated embryos display an average of 20.3 -/+ 6.7 protrusions per hour, while cardiomyocytes in LY-treated embryos display an average of 17.00 -/+ 7.4 protrusions per hour (p-value: 0.3562, Two-tailed t-test), indicating no significant difference between the frequency of protrusions. (This data is now included in our manuscript, page 14 lines 11-16 and in Figure 4-source data.)

5. Figure 5F. The results of 10µM LY inhibitor in Pdgfra +/+ embryos do not match well with Supplementary Figure 1M. Please check and discuss if necessary.

We appreciate the reviewer’s careful reading of our manuscript. In Figure 5F and Figure 1 —figure supplement 1M, separate experiments were conducted involving 10 µM inhibitor treatments. Although, there are slight differences in percent of embryos displaying cardiac phenotypes (None in Figure 1 —figure supplement 1M, compared to an average of 10.9% in Figure 5C), these differences are not statistically significant (pvalue: 0.2784, Two-tailed t-test). The minor variation from these distinct experiments could have arisen from the different genetic backgrounds of the fish used in this experiment or from differences in the batches of inhibitor.

6. Do the authors observe a genetic interaction between dnPI3K and Pdgfras16 mutants (het or homo)?

We appreciate the reviewer’s suggestion, unfortunately this is currently a technically difficult experiment for us to perform. We have however found that we can rescue the cardiac fusion phenotype caused by increased Pdgfra signaling by a sub-phenotypic dose of PI3K inhibitor (Figure 5G-J), further supporting our finding that Pdgfra and PI3K signaling interact to regulate the process of cardiac fusion.

7. Figure 5. Western blots for p-AKT are based on total embryo lysates. Thus, it is expected to find what the authors found related to Pdgfra mutants and hs:pdgfaa. The manuscript would be strengthened if the authors could provide direct evidence of phosphorylated AKT in the myocardial region during cardiac fusion. The authors could use immunofluorescence and confocal imaging.

We thank the reviewer for their suggestion. As we have indicated in our response to

recommendations related to Figures and related conclusions: Query 1, we have

now added results of PH-domain reporter of PI3K activity, which complements our other tissue-specific studies indicating PI3K activity in the myocardium during cardiac fusion.

8. I am a bit conflicted with Supplementary Figure 5C. What each dot represents? A single cell? I suggest the authors read the latest thread about some important statistical considerations:https://doi.org/10.1083/jcb.202001064https://rupress.org/jcb/article/219/6/e202001064/151717/SuperPlots-Communicating-reproducibility-and

We thank the reviewer for their suggestion. We have changed our graphs analyzing cardiomyocyte behavior to super plots – which show variation between individual cells as well as variation between embryos.

Reviewer #3 (Recommendations for the authors):Overall, the manuscript is well written and experiments are well controlled providing sufficient evidence to substantiate most of their conclusions. There are a number of questions that need to be addressed by the authors.Specific points:1. What is interesting is that the CPCs have some intrinsic migration that is independent of the endoderm. The finding that Pi3K is required for the cardiac fusion process is not too surprising as the authors showed in their previous work that PDGF signaling is driving this process and PI3K is a well-known transducer of PDGF signaling. An important question remaining is what type of cell migration the CPCs undergo to fuse at the midline. Are the CPCs migrating as individual cells that all respond to the PDGF from the midline or do the CPCs migrate collectively and only a few respond to the PDGF signal? The authors seem to prefer the collective migration of CPCs but it seems that this has not been tested experimentally. Testing this directly would significantly increase the impact of this study. With the myl7:dnPI3K construct the authors seem to have a good tool to test this now. This could be addressed by analysing embryos in which the cardiac field contains a mixture of wt CPCs and CPCs expressing the dnPIK3 for example by cell transplantation experiments in which cells from embryos derived from the myl7:dnPI3K F1 fish are transplanted to wt embryos or by DNA injections resulting in mosaic expression. It would be very interesting to see how CPCs expressing the dnPI3K behave when surrounded by wt cells. If collective migration is indeed driving cardiac fusion one would expect that these dnPI3K cells behave like wt cells.

We appreciate the reviewer’s suggestion. Cardiomyocytes are interconnected to one another via intercellular junctions starting at around 12s (Ye D, et al. 2015, PMID: 26329600), which would support a collective mode of migration. However, the reviewer is correct that this is still an open question, which we hope to address in a future study.

2. In Figure 3 the authors show that migration of CPCs treated with LY is affected. They plotted the CPC velocities in 3E and here it looks as if fast-migrating CPCs are less affected compared to slower-migrating CPCs. Fast migration is mostly seen in the posterior part of the bilateral heart fields, so the question arises whether CPCs along the anterior-posterior axis is affected differently by the LY treatment.

To address this question, we analyzed cardiomyocyte velocity after dividing the bilateral cardiac populations into thirds: Top (anterior), Middle, and bottom (posterior). However, we did not find a significant difference in the cardiomyocyte velocity between DMSO and PI3K-inhibited embryos. This new information is contained (Figure 3 —figure supplement 2F) and on page 13, lines 8-9.

3. In fig3E the authors show the total velocity of the CPCs in all directions. As they conclude at the end that PDGF-PI3K signaling mostly affects the medial cell migration, it would be important to analyse the velocities of the CPCs only in the medial-lateral direction.

We have now analyzed cardiomyocyte velocity in the medial-lateral direction, where we found a significant difference between DMSO (0.3548 -/+ 0.027 microns/min) and PI3Kinhibited embryos (0.1716 -/+ 0.018 microns/min) (Two-tailed T-test: 0.0008). This information is now found in Figure 3 —figure supplement 2E and on page 13, lines 5-9.

4. In Figure 3G the authors show the angles at which the CPCs migrate based on the time lapses. In Figure 3B the tracks show that some cells (mostly located anteriorly) move in a medial-posterior direction while other cells (mostly located in the posterior part) move in an anterior-medial direction. The rose plot however displays only positive angles (medial-anterior direction) and no angles in the medial-posterior direction. Please explain why this was done.

Our goal when calculating the angle of movement was to determine whether cells were directed along the medial-lateral or were directed angularly (along the anterior-posterior axis, as determined by Holtzman et al. 2007). In this analysis we do not distinguish movement in the anterior direction from movement in the posterior direction, which would be problematic for calculating an average. This is in line with how we calculated directionality in the original *eLife* paper 2017. Unfortunately, this was not clear in our first version, and we thank the reviewer for bringing this to our attention. We have modified the methods to clearly indicate how directionality is calculated (page 31, line 3).

In addition, the angles shown here for wt embryos (average 30 degrees) seem very different from the angles reported for wt embryos in their previous work (Fig6L, average around 60 degrees), which look more like the angles reported here for the LY treated embryos (average around 60 degrees). Please explain this.

We thank the reviewer for their thoughtful consideration of our data. As we indicate in our response to Essential revisions #2 – Overall, our data is consistent with both our previous work and with the work of others (Holtzman et al. 2007). The difference between the two studies may be a result of normalization differences or in slight variations in the developmental stages captured. We now include a discussion of these differences in manuscript (page 18, lines 16-20).

5. In Figure 4 the authors analyse the direction and persistence of cellular protrusion in CPCs. Especially the direction of these protrusions seems affected by LY suggesting that PDGF from the midline may be responsible for this phenomenon. It would therefore be very interesting to test this in their hs:pdgf line by analysing protrusion direction when pdgf is produced by all cells. If PDGF from the midline is responsible for their direction, overexpressing pdgf in all cells should give a similar effect as inhibiting PI3K.

We agree with the reviewer that this is a valuable experiment. Our current manuscript revolves around the role of PI3K signaling in cardiac fusion. As we turn our attention directly to the role of protrusions in cardiac fusion, we think this will be an important experiment to complete.